# Validation of robust radiobiological optimization algorithms based on the mixed beam model for intensity-modulated carbon-ion therapy

**Masashi Yagi**[1,2]*, **Toshiro Tsubouchi**[2], **Noriaki Hamatani**[2], **Masaaki Takashina**[2], **Naoto Saruwatari**[3], **Kazumasa Minami**[3], **Yushi Wakisaka**[4], **Shinichiro Fujitaka**[5], **Shusuke Hirayama**[5], **Hideaki Nihongi**[6], **Azusa Hasegawa**[7], **Masahiko Koizumi**[3], **Shinichi Shimizu**[1], **Kazuhiko Ogawa**[8], **Tatsuaki Kanai**[2]

1 Department of Carbon Ion Radiotherapy, Osaka University Graduate School of Medicine, Suita-shi, Osaka, Japan, 2 Department of Medical Physics, Osaka Heavy Ion Therapy Center, Osaka-shi, Osaka, Japan, 3 Department of Medical Physics and Engineering, Osaka University Graduate School of Medicine, Suita-shi, Osaka, Japan, 4 Department of Radiation Technology, Osaka Heavy Ion Therapy Center, Osaka-shi, Osaka, Japan, 5 Hitachi, Ltd., Research & Development Group, Hitachi-shi, Ibaraki, Japan, 6 Hitachi, Ltd., Healthcare Innovation Division/Healthcare Business Division, Kashiwa-shi, Chiba, Japan, 7 Department of Radiation Oncology, Osaka Heavy Ion Therapy Center, Osaka-shi, Osaka, Japan, 8 Department of Radiation Oncology, Osaka University Graduate School of Medicine, Suita-shi, Osaka, Japan

* m.yagi@radonc.med.osaka-u.ac.jp

**Data Availability Statement:** All relevant data are within the paper.

## Abstract

Currently, treatment planning systems (TPSs) that can compute the intensities of intensity-modulated carbon-ion therapy (IMCT) using scanned carbon-ion beams are limited. In the present study, the computational efficacy of the newly designed IMCT algorithms was analyzed for the first time based on the mixed beam model with respect to the physical and biological doses; moreover, the validity and effectiveness of the robust radiobiological optimization were verified. A dose calculation engine was independently generated to validate a clinical dose determined in the TPS. A biological assay was performed using the HSGc-C5 cell line to validate the calculated surviving fraction (SF). Both spot control (SC) and voxel-wise worst-case scenario (WC) algorithms were employed for robust radiobiological optimization followed by their application in a Radiation Therapy Oncology Group benchmark phantom under homogeneous and heterogeneous conditions and a clinical case for range and position errors. Importantly, for the first time, both SC and WC algorithms were implemented in the integrated TPS platform that can compute the intensities of IMCT using scanned carbon-ion beams for robust radiobiological optimization. For assessing the robustness, the difference between the maximum and minimum values of a dose–volume histogram index in the examined error scenarios was considered as a robustness index. The relative biological effectiveness (RBE) determined by the independent dose calculation engine exhibited a −0.6% difference compared with the RBE defined by the TPS at the isocenter, whereas the measured and the calculated SF were similar. Regardless of the objects, compared with the conventional IMCT, the robust radiobiological optimization enhanced the sensitivity of the examined error scenarios by up to 19% for the robustness

**Funding:** This work was partially supported by the JSPS KAKENHI 17K16437, 21K07700, and 22K07695.

**Competing interests:** The authors, Shinichiro Fujitaka, Shusuke Hirayama, and Hideaki Nihongi are an employee of Hitachi, Ltd.

index. The computational efficacy of the novel IMCT algorithms was verified according to the mixed beam model with respect to the physical and biological doses. The robust radiobiological optimizations lowered the impact of range and position uncertainties considerably in the examined scenarios. The robustness of the WC algorithm was more enhanced compared with that of the SC algorithm. Nevertheless, the SC algorithm can be used as an alternative to the WC IMCT algorithm with respect to the computational cost.

## Introduction

Heavy-charged particles—particularly carbon ions—have attracted increasing attention as potential cancer therapy candidates [1]. These ions exhibit promising characteristics, such as a beneficial depth–dose profile, minimal lateral scattering, and an enhanced biological efficacy across the Bragg peak. Pencil beam scanning in 3D enhances these characteristics for particle therapy.

The intensity modulation method can be used to additionally enhance carbon-ion therapy by delivering highly conformal dose distributions in tumors with complex shapes and averting undesired exposure to adjacent organs at risk (OAR). Various terms are used to describe these intensity modulation techniques for carbon ions: intensity-modulated particle therapy (IMPT), intensity-modulated ion therapy (IMIT), and intensity-modulated carbon-ion therapy (IMCT). In the current study, the term IMCT is used when IMPT is specifically indicated with the carbon-ion beam.

In IMCT, non-uniform dose distributions, commonly with strong in-field dose gradients, are delivered from various directions; the prescribed distributions are acquired after superposing the dose contributions from all fields. This is different from conventional carbon-ion therapy using a single-field uniform dose (SFUD). Therefore, compared with the conventional carbon-ion therapy using SFUD, IMCT is possibly more susceptible to range and setup uncertainties [2, 3]. There is thus a requirement for assessing the robustness of IMCT plans [4, 5]. For IMCT, both the physical and biological doses should be considered for robust optimization and dose computation using a scanned carbon-ion beam.

The calculation of the IMCT treatment plan was accomplished using the treatment planning system (TPS) VQA Plan (Hitachi, Ltd., Tokyo, Japan) [6–10]. The IMCT functions in VQA Plan, including robust optimization and dose calculation, have recently been generated for the carbon-ion beam at the Osaka Heavy Ion Therapy Center (OHITC). The TPS determines the physical dose using an analytical dose calculation algorithm, a pencil beam [11] model with a triple Gaussian form [12] for the lateral dose distribution, and a beam splitting algorithm [13] to consider lateral heterogeneity in the medium. For determining the relative biological effectiveness (RBE) of the IMCT, we adopted the mixed beam model for the first time [14] as the RBE model. Here the "mixed" means a mixed linear-energy transfer (LET). Both spot control (SC) and voxel-wise worst-case scenario (WC) algorithms were used for robust radiobiological optimization. These algorithms were never used in an integrated TPS platform.

Modern robust optimization algorithms incorporate uncertainties including range and/or setup [15–18] and organ motion [19, 20] directly in the optimization algorithm; these approaches considerably reduce the sensitivity of delivered dose distributions to these uncertainties. The SC algorithm [21] has been designed for OAR sparing because range and/or setup errors can induce an adverse dose burden in adjacent OARs. In this algorithm, only a nominal scenario was analyzed in the iterative optimization process by incorporating the term

"penalizing the pencil beam with a risk of delivering a high dose to the OARs" into the objective function. The SC algorithm was used with the in-field dose gradient suppression function for increasing its robustness of range and/or setup errors. Voxel-wise WC algorithm [16] is a part of the minimax approaches that considers an extreme case in the minimax stochastic programming for robust optimization. The algorithm aims to acquire an optimal treatment plan for the worst error scenario that is separately considered for each voxel.

Currently, commercially available TPSs that can compute the intensities of IMCT using scanned carbon-ion beams are limited. Monaco-I (Elekta AB, Stockholm, Kingdom of Sweden), which is a commercial version of the XiDose system developed at the Japan Agency for Quantum and Radiological Science and Technology, uses the SC algorithm for robust radiobiological optimization. This system uses the microdosimetric kinetic (MK) model [22] as the RBE model. RayStation (RaySearch Laboratories AB, Stockholm, Kingdom of Sweden) is another TPS used for carbon-ion treatment planning; it has implemented the composite WC algorithm for robust radiobiological optimization and is a part of the minimax approaches aimed at obtaining the treatment plan that is best possible for the worst error scenario considered. When only the VQA Plan implemented both the robust radiobiological optimization algorithms, the local effect model I [23] or the MK model can be selected as the RBE model.

In the current study, the computational efficacy of novel IMCT algorithms based on the mixed beam model was examined; physical and biological doses were applied to an Radiation Therapy Oncology Group (RTOG) benchmark phantom under both homogeneous and heterogeneous conditions as well as a clinical case was used to verify the validity and efficacy of the algorithms in terms of the robust radiobiological optimization.

## Materials and methods

### IMCT in VQA plan

The TPS used in the present study was the VQA Plan version 5.10 (Hitachi, Ltd.) that included a carbon module for scanned carbon-ion beam delivery. To calculate the RBE of the IMCT, the mixed beam model [14] was used as the RBE model. SC and WC algorithms were employed for robust radiobiological optimization.

Previous studies have provided a detailed discussion of the dosage calculation method and the validity used for non-IMCT such as SFUD [6, 9]. In the present study, the difference in dose calculation between non-IMCT and IMCT was described. Furthermore, the physical dose at the calculation spot $i$—$d_i$ in IMCT—was provided with respect to the dose contributions from distinct fields comprising the IMCT independently as follows:

$$d_i(x) = \sum_{j=1}^{N} d_{ij} x_j \tag{1}$$

where $d_{ij}$ is the dose contribution of beam $j$ at the spot $i$ from different fields involved in the IMCT, and $x_j$ is the number of particles in carbon-ion beam $j$ in MU.

In the mixed beam model, the RBE was calculated using the survival curves of human salivary gland (HSG) cells for photons and carbon ions by employing a specified survival level (10%) based on the linear quadratic model (LQM). The biological dose was thus determined by the following formula:

$$d_{bio,i} = \frac{\sqrt{\alpha_X^2 + 4\beta_X e_i} - \alpha_X}{2\beta_X} \tag{2}$$

where $\alpha_X$ and $\beta_X$ are the LQM parameters for photons. Furthermore, $e_i$, which represents the

biological effect at spot $i$, was represented by the following formula:

$$e_i = \alpha_i d_i + \beta_i d_i^2$$

where $\alpha_i$ and $\beta_i$ are the LQM parameters for carbon ions that were calculated with respect to the dose contributions from different fields involving the IMCT independently as follows:

$$\alpha_i = \frac{1}{d_i} \sum_{j=1}^{N} \alpha_{ij} d_{ij} x_j \tag{3}$$

$$\sqrt{\beta_i} = \frac{1}{d_i} \sum_{j=1}^{N} \sqrt{\beta_{ij}} d_{ij} x_j \tag{4}$$

where $\alpha_{ij}$ and $\beta_{ij}$ are the LQM parameters for carbon ions that were determined according to the contributions of beam $j$ at the spot $i$ from different fields comprising the IMCT. These LQM parameters are functions of the LET and are obtained from [6].

**SC algorithm.** The optimization for IMCT can be summarized using the following formula:

$$\underset{x \geq 0}{\text{minimize}} \ F_k(x) \tag{5}$$

where $F_k$ denotes the objective function of the robustness term $k$, and $x$ is the vector of beam weights.

The objective function of the SC algorithm comprised the following three terms: 1. $F_{hetero}(x_j)$, which denotes dose heterogeneity suppression in a target; 2. $F_{grad}(x_j)$ which denotes in-field dose gradient suppression; and 3. $F_{OAR\_sparing}(x_j)$, which denotes dose heterogeneity suppression in an OAR in addition to the objective function for the nominal dose distribution $D_{nom}$.

$$F_{SC}\left(x_j\right) = F\left(D_{nom}\left(x_j\right)\right) + F_{hetero}\left(x_j\right) + F_{grad}\left(x_j\right) + F_{OAR\_sparing}\left(x_j\right) \tag{6}$$

Tissue heterogeneities lateral to the beam direction lead to high sensitivity in setup errors. A density index [24, 25] $m$ was introduced to quantify lateral tissue heterogeneities of a single beam $j$ at the spot $i$.

$$m_j = \sum_{i \in A} \left(L_j - L_i\right)^2 \tag{7}$$

where $L_j$ and $L_i$ are the water-equivalent-length from the source to a range depth of the beam $j$ and spot $i$ in the calculation area $A$, respectively.

The following formula was used to determine dose heterogeneity suppression in a target:

$$F_{hetero}\left(x_j\right) = K_0 \frac{\sum_{j=1}^{N} m_j x_j}{\sum_{j=1}^{N} x_j} \tag{8}$$

where $K_0$ is the penalty coefficient in the optimization. Before the present study, the combination of the dose heterogeneity suppression term with the other terms had never been investigated.

Loose longitudinal and lateral dose gradients render treatment plans insensitive to range and setup errors, respectively [26]. In-field dose gradient suppression was expressed using

the following formula:

$$F_{grad}\left(x_j\right) = G_0 \sum_{l=1}^{N_F} \sum_{i \in T} \sum_{\substack{k \in T}} \left(\frac{e_{l,i} - e_{l,k}}{r_{ik}}\right)^2 \tag{9}$$
$$i \neq k, r_{ik} \leq \Delta r$$

where $G_0$ is the penalty coefficient in the optimization, $N_F$ represents the number of fields comprising the IMCT, $e_{l,i}$, $e_{l,k}$, and $r_{ik}$ are the biological effect of field $l$ at the spots $i$ and $k$ and the distance between the spots, respectively, whereas $\Delta r$ indicates a calculation distance of interest. This equation can facilitate the reduction of the variation of the biological effect within the $\Delta r$.

Typically, only a small proportion of pencil beams are at a risk of delivering a high dose to surrounding OARs when range and setup uncertainties arise in a treatment plan. To compensate for these pencil beams, the risk index $P_j$ that quantifies the risk for a single pencil beam $j$ was introduced [21] into the dose optimization by adding the following terms to the objective function:

$$F_{OAR-sparing}\left(x_j\right) = \frac{\Sigma_{j=1}^{N} x_j \times \left(K_1 P_j^R + K_2 P_j^S\right)}{\Sigma_{j=1}^{N} x_j} \tag{10}$$

where $K_1$ and $K_2$ are the penalty coefficients of the risk index for range uncertainty, $P_j^R$, and setup uncertainty, $P_j^S$, respectively in the optimization. In the present study, the risk index was simply calculated using geometrical information in the algorithm created by our group instead of dose calculation derived alternatively in a prior study [21]. Moreover, the $P_j^R$ was increased when the Bragg peak was close to the OAR in an error scenario of range error, whereas the $P_j^S$ was increased when the beam path was close to the OAR in an error scenario of setup error.

**WC algorithm.** In the voxel-wise WC algorithm [16], each voxel is considered to be independently influenced by the uncertainty, and the penalty to each voxel depends on the worst dose that the voxel can receive under the examined errors. The physical dose at the spot $i$ in an error scenario $d_i^s$ was determined using the following formula:

$$d_i^s(x; s) = \sum_{j=1}^{N} d_{ij}^s x_j \tag{11}$$

where $d_{ij}^s$ is the dose contribution of beam $j$ at the spot $i$ from different fields including the IMCT in the error scenario $s$, whereas $e_i^s$, which describes the biological effect at the spot $i$ in an error scenario $s$ was represented by the following formula:

$$e_i^s = \alpha_i^s d_i^s + \beta_i^s d_i^{s^2} \tag{12}$$

where $\alpha_i^s$ and $\beta_i^s$ are the LQM parameters for carbon ions that were calculated according to the dose contributions from different fields involved in the IMCT in the error scenario $s$.

The objective function of the WC algorithm comprised the following two terms:

$$F_{WC}(x) = F(D_{nom}(x)) + p_w F(D_w(x)) \tag{13}$$

where $p_w$ is the importance of the WC dose distribution $D_w$. The objective function of $D_w$ was

determined using the following formula:

$$F(D_w(x)) = \sum_{i \in T} w_i f_{w,\,target}\,(x) + \sum_{i \in O} w_i f_{w,OAR}(x) \tag{14}$$

$$f_{w,\,target}\,(x) = \sum_{i=1}^{m} \left\{ \left( \min_s e_i^s - e_i \right)^2 + \left( \max_s e_i^s - e_i \right)^2 \right\} \tag{15}$$

$$f_{w,OAR}(x) = \sum_{i=1}^{m} \left\{ \theta \left( \max_s e_i^s - e_{max} \right) \left( \max_s e_i^s - e_{max} \right)^2 \right\} \tag{16}$$

where $\theta$, $\min_s e_i^s$ and $\max_s e_i^s$ are the step function and the minimum and maximum biological effects at the spot $i$ in all error scenarios, respectively.

## Validation

**Physical approach.** A C-shaped target resembling CTV (inner diameter: 18 mm, outer diameter: 40 mm, length: 40 mm) and a cylindrical OAR (diameter: 30 mm, length: 40 mm) were generated in a virtual water phantom of $100 \times 100 \times 100$ mm$^3$. Relative stopping power (RSP) was set to 1.0. The CTV with three fields of a horizontal port comprising a couch rotation of 0˚, 180˚, and 270˚ in the TPS provided a dose of 5.88 Gy(RBE) that yielded 10% of the survival rate of HSG cells. This treatment plan was generated without conducting radiobiological optimization.

A dose calculation engine was independently developed using Python (version 3.6.10) to validate a clinical dose calculated in the TPS. The treatment plan data (beam energy, spot position, and MU) were retrieved from the TPS and used to calculate the clinical dose in the in-house dose calculation engine. For a dose comparison between the TPS and the in-house software, a 3D local gamma index analysis using 2 mm/3%, with a 10% threshold, was conducted. For this assessment, the VeriSoft software version 7.1 (PTW-Freiburg GmbH, Freiburg, Germany) was used.

**Biological approach.** The biological dose calculation validation can compare the RBE values determined using the newly established TPS with the RBE values determined using a validated TPS that applies the same RBE model concept as the newly designed TPS [9]. However, the mixed beam model was adopted as the RBE model for determining the RBE of the IMCT for the first time. RBE of the IMCT can only be computed with a cell experiment.

For the purposes of this study, a dedicated cubic water phantom (Taisei Medical Inc., Osaka, Japan) was developed for the described biological experiment. Initially, a treatment plan was designed for a donut-shaped target (inner diameter: 25 mm, outer diameter: 100 mm, length: 100 mm) in the cubic water phantom [Fig 1(A)–1(C)]. Regarding three-beam arrangements, the robust radiobiologically optimized treatment plan with WC was computed including 14 error scenarios (7 per caron-ion range), the calculated minimum (−3.5%) and maximum (+3.5%) carbon-ion ranges, and nominal and 2-mm shifted positions in anteroposterior (AP), superoinferior (SI), and left–right directions (LR) [27, 28]. The prescribed dose was 5.88 Gy(RBE) leading to a 10% surviving fraction (SF) of HSG with the mixed beam model to the target, whereas the dose in the hole of the donut-shaped target resembling an OAR was optimally decreased. Using similar beam arrangements and the prescription dose, the SFUD plan was also designed using one field with a couch angle of 0˚ for comparison purposes. In this case, an OAR could not be prevented solely to examine the SF of the target.

(a)

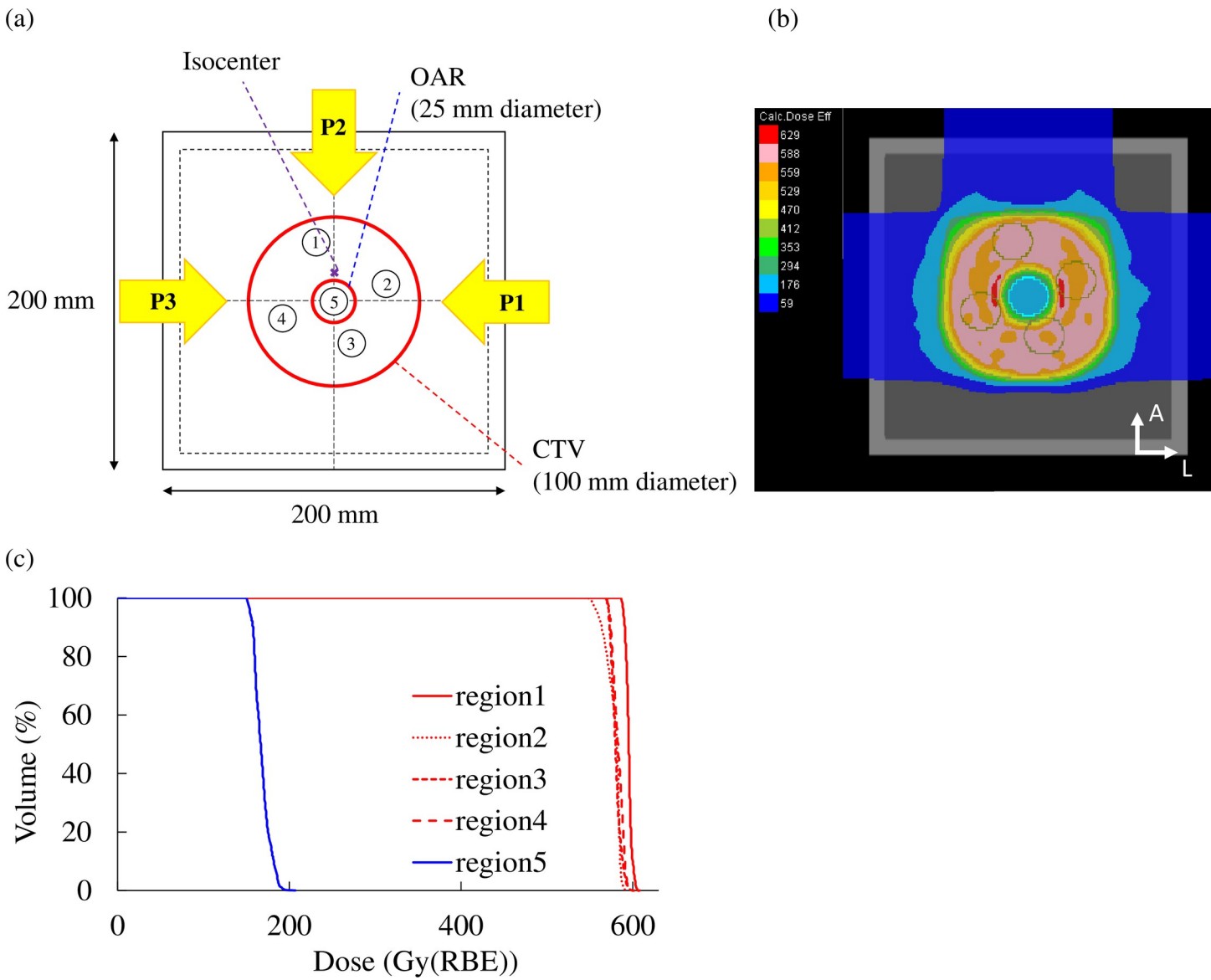

(b)

(c)

**Fig 1. Treatment plan for the biological study.** (a) Beam arrangement to a donut-shaped target. Dose distribution (b) and dose–volume histogram (c) of the treatment plan.

The generated treatment plan was irradiated with HyBeat Heavy ion Therapy System (Hitachi, Ltd.) to the cell flasks inside the water phantom. Four of the examined flasks were set within the target region and one within the OAR region [Fig 1(A)–1(C)]. The center of the water phantom was aligned to the isocenter using a room laser followed by an imaging system called PIAS (Patient Positioning Image and Analysis System). Physical doses were measured before irradiating cells using a PinPoint 3D chamber (type 31016, PTW-Freiburg GmbH) attached to an electrometer (UNIDOS webline, PTW-Freiburg GmbH) in each flask.

HSG (HSGc-C5) cells were cultured in Dulbecco's Modified Eagle's Medium–high glucose (Sigma Aldrich, Burlington, MA, United States) supplemented with 10% fetal bovine serum (Gibco, Thermo Fisher Scientific, Waltham, MA, United States) and 1% Penicillin–Streptomycin–Glutamine Mixed Solution (Nacalai Tesque, Kyoto, Japan). Cells were sustained at 37°C

and 5% $CO_2$. After carbon-ion irradiation, 500 or 1000 cells were seeded in 6 cm φ cell culture dishes (Corning, Somerville, MA, US), with three dishes per one condition. These dishes were incubated for 2 weeks. Then, the cells were fixed with formaldehyde (10% solution, Sigma Aldrich) for 20 min before staining with crystal violet (FUJIFILM Wako Pure Chemical Corporation, Osaka, Japan). The dishes were then washed with water. When a cell population reached a size of ≥50 cells, it was considered a colony [29].

## Robustness evaluation

A dose of 4.8 Gy(RBE) was recommended for an RTOG benchmark phantom geometry where a C-shaped CTV (inner diameter: 18 mm, outer diameter: 40 mm, length: 40 mm) surrounds a cylindrical OAR (diameter: 30 mm, length: 40 mm) in both homogeneous and heterogeneous phantoms (diameter: 232 mm, length: 80 mm) using a three-field IMCT with the port angles of 0˚, 90˚, and 270˚. The dose for OAR was lowered to the lowest possible level. The heterogeneous phantom comprises air and bone regions adjacent to the CTV (Fig 2). RSPs of air and bone was set to 0.001 and 1.53, respectively. A geometrical margin of 2 mm that was applied in the clinic was added in the CTV but not for the evaluation of the WC algorithm.

Robustness evaluation of the generated plan was attained by applying the "Analysis" function of the TPS where dose calculations were performed against each specified error scenario. For robustness evaluation, the combinations of range and setup uncertainties were considered (in total 28 uncertainty scenarios), including the minimum (+3.5%) and maximum (−3.5%) carbon-ion ranges and the combination of 2-mm shifted positions in AP, SI, and LR directions (14 per carbon-ion range). The difference between the maximum and minimum of $D_{98}$ in the

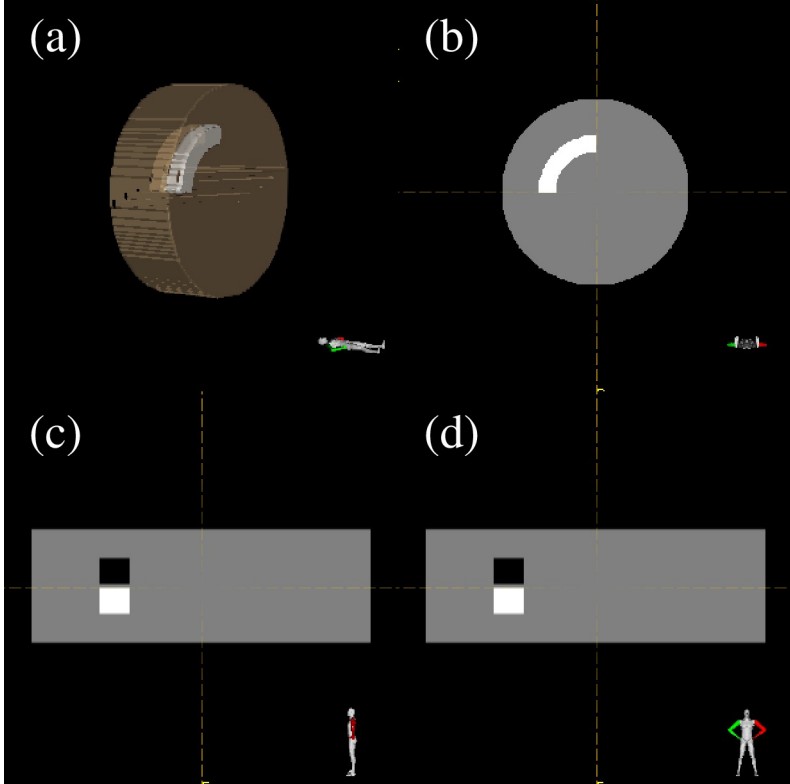

**Fig 2. The heterogeneous phantom.** Three-dimension (a), axial (b), sagittal (b), and coronal (d) are depicted.

studied scenarios (robustness index) was analyzed for the robustness of the CTV, whereas that of $D_{max}^{OAR}$ in the evaluated scenarios was examined for the robustness of the OAR.

**Robustness evaluation for SC algorithm.** Three different treatment plans for both homogeneous and heterogeneous phantoms were performed to evaluate the robust radiobiological optimization algorithm against range and setup uncertainties. A prior investigation demonstrated that although there were several combinations of robust optimization terms in the SC algorithm, more penalties should be considered for range uncertainties than setup uncertainties to minimize the WC OAR dose and maintain the target coverage [21]. Robust radiobiological optimization was performed using higher penalties placed on the range than setup uncertainties combined with heterogeneity or gradient suppression. Table 1 summarizes the details of the optimization parameters.

To examine the efficacy of the SC algorithm for reducing the sensitivity of IMCT plans against range and setup uncertainties, three IMCT plans were compared:

- A conventional IMCT plan that does not involve any uncertainty (plan-1: a conventional IMCT plan),

- A plan that takes into account both the target coverage based on heterogeneity suppression and OAR sparing to analyze range and setup uncertainties with an increased penalty for range uncertainties (plan-2: a range, setup, and heterogeneity robust IMCT plan) and

- A plan that evaluates both target coverage based on gradient suppression and OAR sparing to account for range and setup uncertainties with an increased penalty for range uncertainties (plan-3: a range, setup, and gradient robust IMCT plan).

**Robustness evaluation for WC algorithm.** Two IMCT plans were compared to analyze the efficacy of the WC algorithm for reducing the sensitivity of IMCT plans against range and setup uncertainties: the conventional IMCT plan similar to the first plan defined in the previous section, and the WC IMCT plan. The combinations of range and setup uncertainty were examined for the robust radiobiological optimization in the WC IMCT plan (in a total of 14 uncertainty scenarios), including the minimum (+3.5%) and maximum (−3.5%) carbon-ion ranges as well as the nominal and the 2-mm shifted positions in AP, SI, and LR directions (7 per carbon-ion range).

## Patient plan

To demonstrate the clinical applicability of the developed algorithm, IMCT plans were generated for a patient with a nasal musculo-epithelial carcinoma surrounding the optic nerve with three beams comprising the port angles of 0˚, 90˚, and 270˚. The prescribed dose within the target was set to 57.6 Gy(RBE). The dose to OAR (i.e., right optic nerve) was decreased to the lowest possible level. The range and setup uncertainties considered were ±3.5% [30] and ±2

**Table 1. Optimization parameters for the three treatment plans performed for the homogeneous and heterogeneous phantoms.**

| No | Plan name | Heterogeneity | OAR sparing | | Gradient | Range uncertainty | Setup error |
|---|---|---|---|---|---|---|---|
| | | | (Range uncertainty) | (Setup error) | | | |
| | | K0 | K1 | K2 | G0 | (%) | (mm) |
| 1 | Conventional IMCT | 0 | 0 | 0 | 0 | 0 | 0 |
| 2 | Range (Setup) +Heterogeneity robust IMCT | 0.5 | 10 | 5 | 0 | 3.5 | 2 |
| 3 | Range (Setup) +Gradient robust IMCT | 0 | 10 | 5 | 0.5 | 3.5 | 2 |

mm [16], respectively. For the SC algorithm, the optimization parameters were similar to that of plan 3 presented in Table 1. Moreover, the WC algorithm included the same scenario described in the previous session (i.e., 14 scenarios) for optimization.

For robustness analysis against range and setup uncertainties, conventional and robust radiobiologically optimized IMCT plans were re-determined with geometrical perturbations. For this analysis, the combinations of range and setup uncertainties were identical as in the case of the phantom study.

All patient records used in this study were approved by the institutional review board of OHITC (IRB, No. 210803). A consent form was waived as this was a retrospective study utilizing de-identified data from patients that have completed carbon-ion therapy.

## Results

### Validation

**Physical approach.** Fig 3 demonstrates that the gamma passing rate was 97.2% for the physical dose and 98.3% for the clinical dose under gamma analysis using 2 mm/3%. Compared with the RBE determined by the TPS at the isocenter, that determined by the independent dose calculation engine revealed a difference of −0.6%.

**Biological approach.** The measured and calculated SF are presented in Fig 4. The measured SF of each flask (1–5) irradiated by IMCT was 23.0%, 26.9%, 31.5%, 25.7%, and 55.6%, respectively. The measured SF of each flask (1–5) irradiated by SFUD was 22.7%, 24.6%, 28.8%, 23.0%, and 24.4%, respectively. The calculated SF of each flask (1–5) irradiated by IMCT was 9.2%, 9.7%, 10.2%, 8.9%, and 52.0%, respectively.

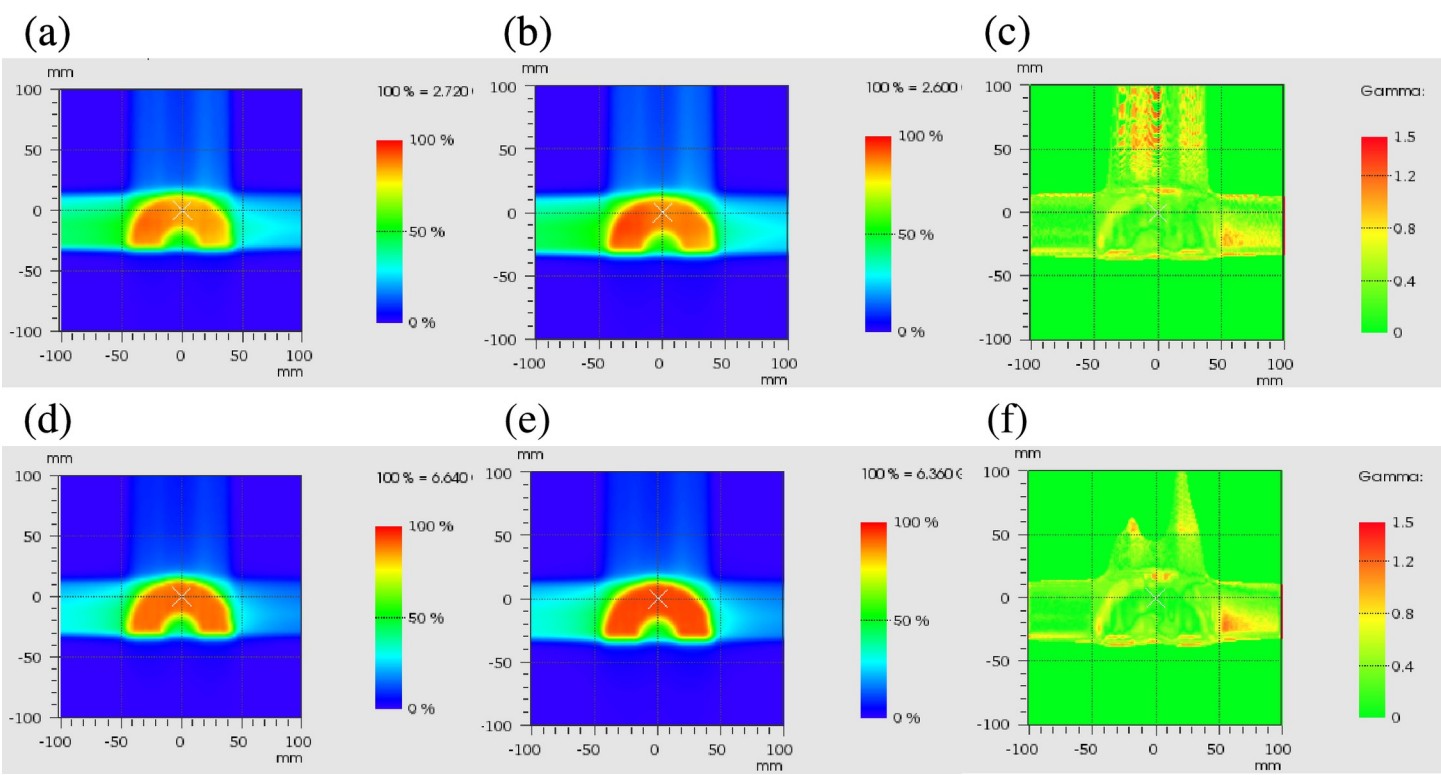

**Fig 3. Dose distribution comparisons.** Physical dose distribution of the independent calculation (a) and TPS (b), and gamma analysis result (c). Clinical dose distribution of the independent calculation (d) and TPS (e), and gamma analysis result (f).

## Robustness evaluation

**Robustness evaluation for SC algorithm.** The representative dose distribution for the nominal case of plans 1 to 3 in the homogeneous phantom, respectively, are shown in Fig 5. Compared with plan-1 [Fig 5(A)–5(E)], as shown in Fig 5(F)–5(H), the beam weight in plan-2 was increased for the beam close to the center of the phantom. The gradient of the physical dose of each beam in plan-3 was mild as presented in Fig 5(K)–5(M). The optimization in plan-3 disallowed a high weight to the Bragg peaks positioned in front of the OAR. To flatten the dose distribution compared with the other plans, the 200-cGy irradiated area was smeared out [Fig 5(N)].

The recalculated perturbed clinical dose distribution with the phantom effective densities of −3.5% or +3.5% and intentional translation of +2 mm in every direction for all fields is presented in Fig 6. The dose distribution of plan-2 [Fig 6(C) and 6(D)] showed the highest deterioration, whereas the dose distribution of plan-3 [Fig 6(E) and 6(F)] demonstrated the least deterioration among the plans owing to a decent sparing of the OAR and a decent CTV coverage.

The dose–volume histograms (DVHs) of the nominal and recalculated perturbed dose distributions are demonstrated with thick solid curves in Fig 7(A)–7(C) and the hatched area in Fig 7(A)–7(C). The robustness indices of the CTV and OAR are presented in Table 2. Compared with plan-1 [Fig 7(A)], the robustness was significantly increased in plan-3 [Fig 7(C)]. Moreover, compared with plan-1, the $\Delta D_{98}$ and $\Delta D_{max}^{OAR}$ were decreased by 4% and 7% for plan-3. In plan-2, the sensitivity against range and setup uncertainties of the OAR was lowered;

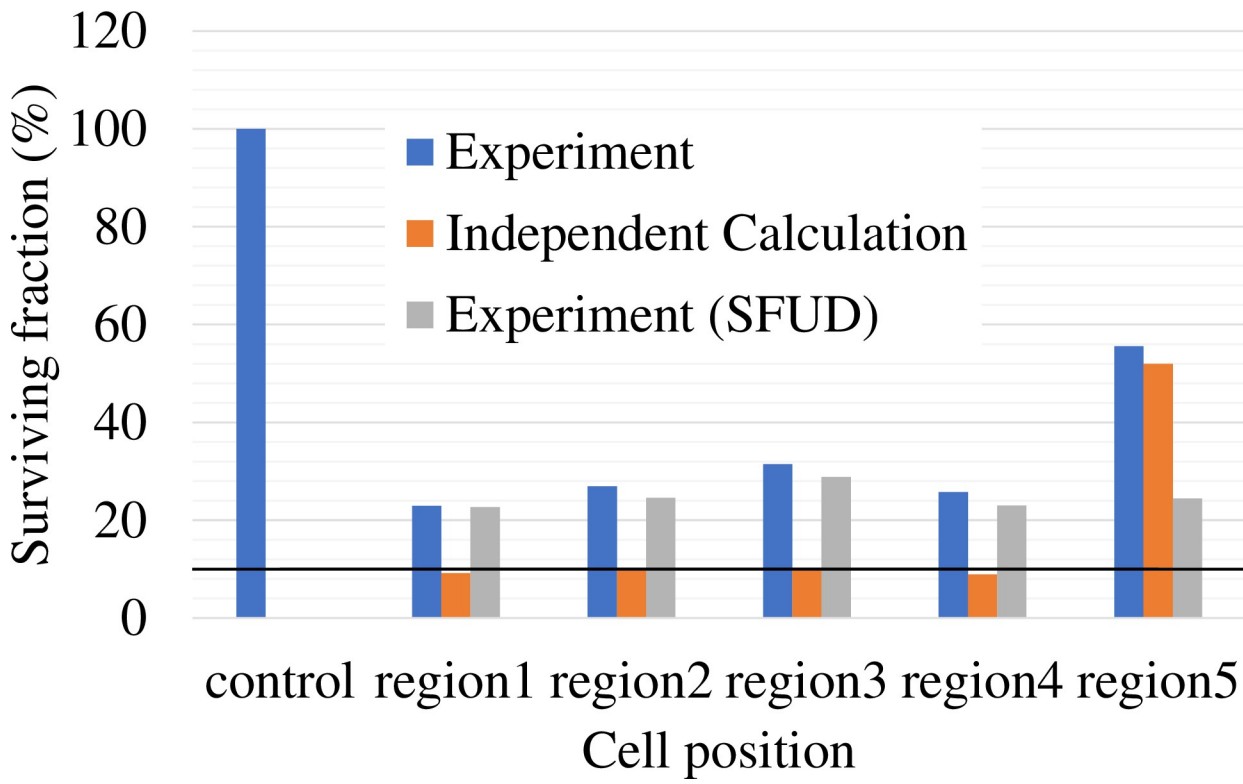

**Fig 4. The SF of HSGc-C5 cells comparison among the experiment irradiated by IMCT (blue bar), the independent calculation (orange bar), and the experiment irradiated by SFUD (gray bar).** The black horizontal solid line represents 10% of SF.

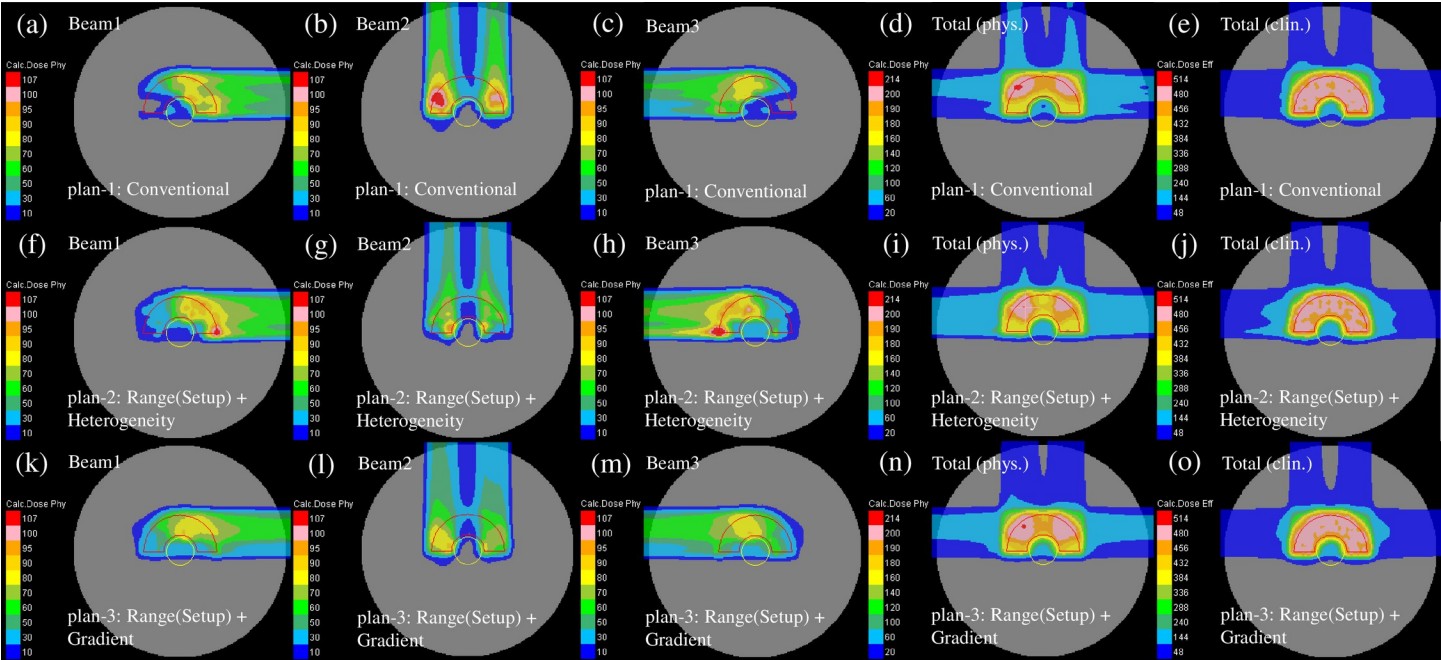

**Fig 5.** Individual dose distributions in the homogeneous phantom of the beams with the port angles of 0˚ (the first column from the left), 90˚ (the second column from the left), and 270˚ (middle column) and the total physical dose distribution (the second column from the right) and clinical dose distribution (the first column from the right) for plan-1 (upper row), plan-2 (middle row) and plan-3 (lower low) optimized with the SC algorithm. The CTV and OAR are represented by the red and yellow lines, respectively.

however, plan-2 was more sensitive for the CTV among the examined plans. The $D_{98}$ in the minimum scenario was the highest and $D_{max}^{OAR}$ in maximum was the lowest for plan-3.

The representative dose distribution for the nominal case of plans-1 to 3 in the heterogeneous phantom is presented in Fig 8. Compared with plan-1 [Fig 8(A)–8(E)], the beam weight in plan-2 was increased for the beam not passing through the heterogeneous region of the phantom [Fig 8(F)–8(H)]. As displayed in Fig 8(K)–8(M), plan-3 revealed dose distributions similar to those of the homogeneous phantom [Fig 5(K)–5(O)].

The recalculated perturbed clinical dose distribution with the phantom effective densities of −3.5% or +3.5% and intentional translation of +2 mm in each direction for all fields is presented in Fig 9. Compared with the homogeneous phantom (Fig 6), the dose distribution of all plans was weakened in the heterogeneous phantom.

The DVHs of the nominal and recalculated perturbed dose distributions are demonstrated as thick solid curves in Fig 7(D)–7(F) and the hatched area in Fig 7(D)–7(F). The robustness indicators of the CTV and OAR are presented in Table 2. The outcomes were comparable to those in homogeneous phantom [Fig 7(A)–7(C) and Table 2].

**Robustness evaluation for WC algorithm.** Fig 10 indicates the representative dose distribution for the nominal case of the conventional IMCT and WC IMCT in the homogeneous phantom. As demonstrated in Fig 10(F)–10(H), the weight of beam 2 was lowered. Compared with the conventional IMCT plan, the weights to the Bragg peaks arranged in front of the OAR were applied to shape the gentle dose gradient [Fig 10(A)–10(C)]. The 200-cGy irradiated area was smeared out to flatten the dose distribution compared with the other plan [Fig 10(I)].

The reassessed perturbed dose distribution with phantom effective densities of −3.5% or +3.5% and intentional translation of +2 mm in each direction for all fields is presented in

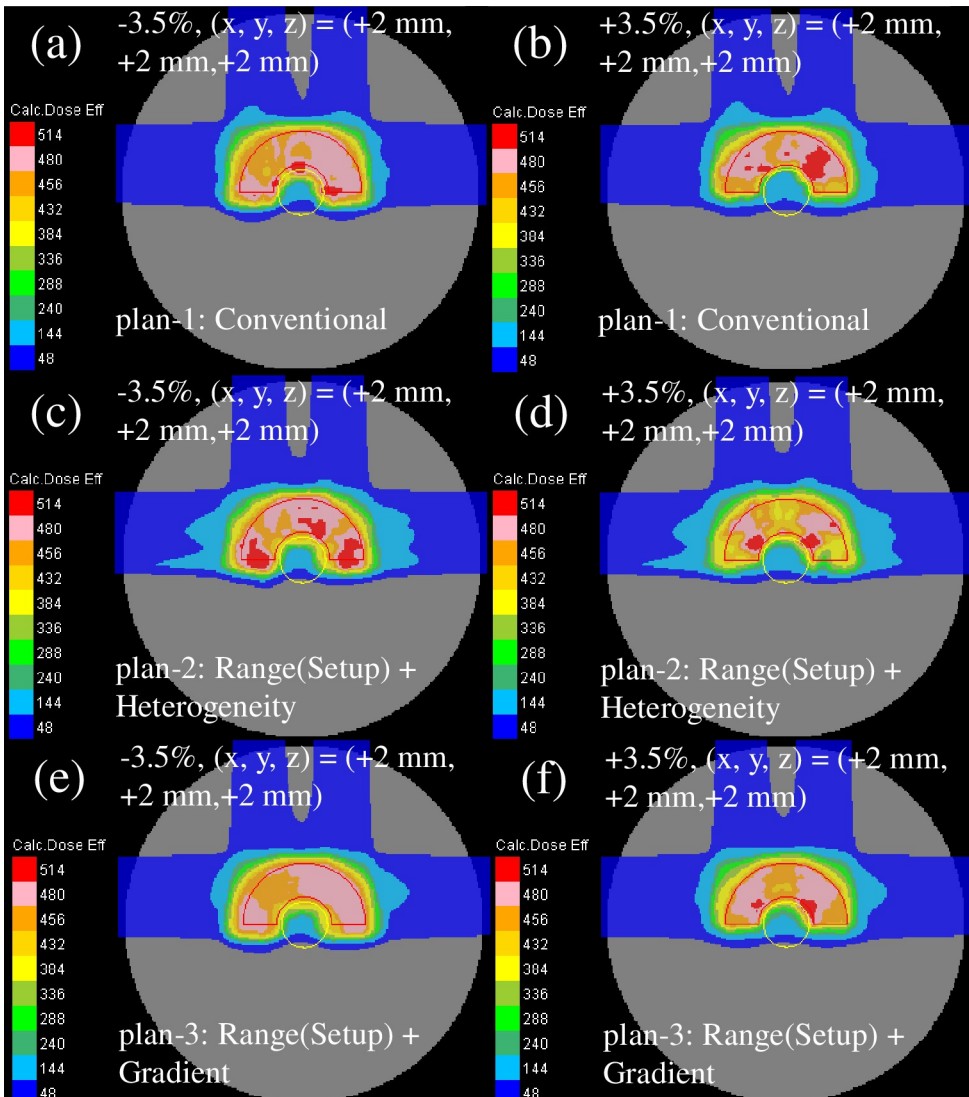

**Fig 6.** Total clinical dose distributions optimized with the SC algorithm in the homogeneous phantom for plan-1 (upper row), plan-2 (middle row), and plan-3 (lower row) recalculated with the efficient density perturbations of −3.5% (left column) and +3.5% (right column) at the intentional translation of +2 mm in each direction for all fields. The red and yellow lines depict the CTV and OAR, respectively.

Fig 11. The dose distribution of the WC IMCT [Fig 11(C) and 11(D)] demonstrated a lower reduction among the plans owing to the decent sparing of the OAR and decent CTV coverage.

The DVHs of the nominal and reassessed perturbed dose distributions are presented with thick solid curves in Fig 12(A) and 12(B) and the hatched area in Fig 12(A) and 12(B). The robustness indices of the CTV and OAR are presented in Table 3. Compared with the conventional IMCT, the $\Delta D_{98}$ and $\Delta D_{max}^{OAR}$ decreased by 11% and 19% for the WC IMCT. The $D_{98}$ in the minimum scenario was the highest and $D_{max}^{OAR}$ in maximum was the lowest for the WC IMCT.

Fig 13 illustrates the representative dose distribution for the nominal case of the conventional IMCT and WC IMCT in the heterogeneous phantom. Compared with conventional

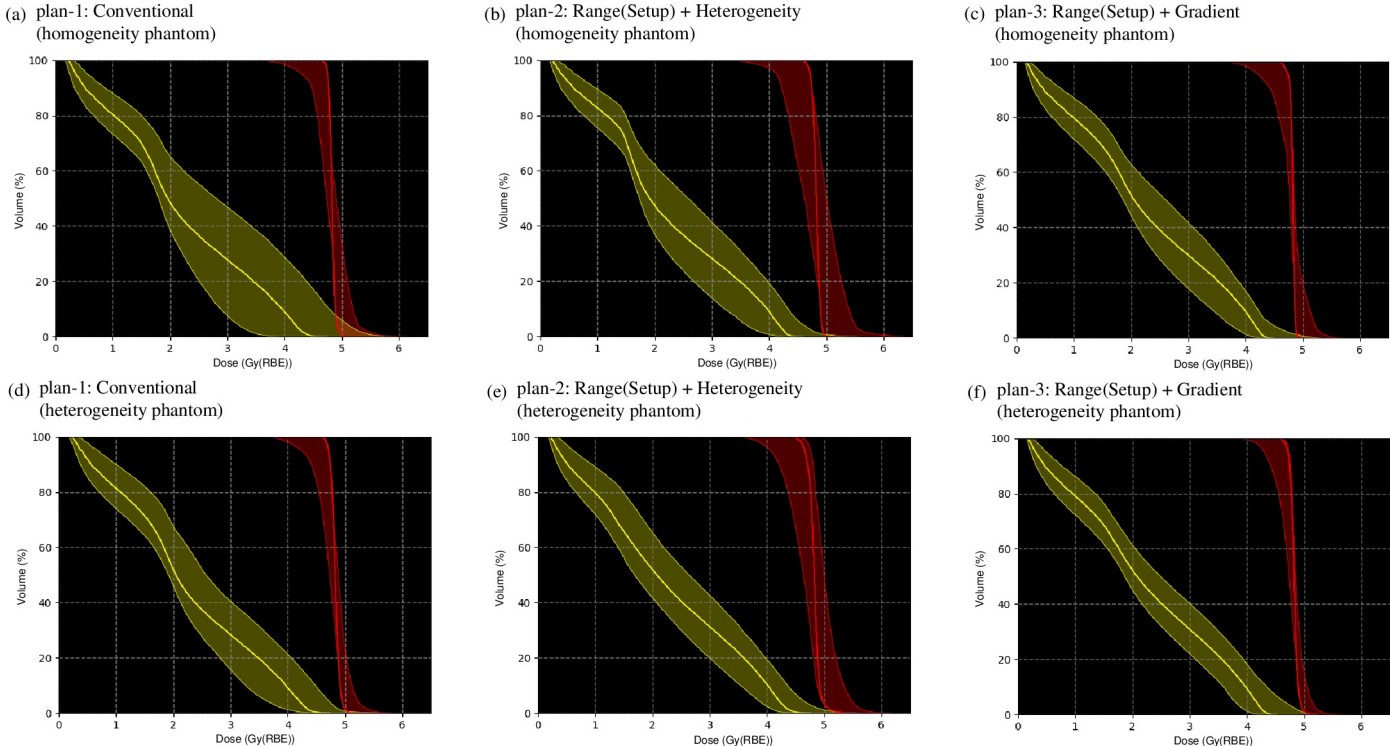

**Fig 7.** The variation of the DVHs of dose distributions optimized using the SC algorithm in homogeneous (upper row) and heterogeneous (lower row) phantoms reassessed for 28 different perturbations of beam ranges and positions in the CTV and OAR for plan-1, plan-2, and plan-3. The DVHs in the nominal case are represented by the thick solid line.

IMCT [Fig 13(A)–13(E)], the weight of beam 1 was higher for the beam not entering the heterogeneous region of the phantom [Fig 13(F)–13(H)].

Fig 14 illustrates the reassessed perturbed dose distribution with phantom effective densities of −3.5% or +3.5% and intentional translation of +2 mm in each direction for all fields.

**Table 2. The $D_{98}$ in the CTV and the $D_{max}^{OAR}$ for the nominal scenario optimized using the SC algorithm in homogeneous and heterogeneous phantoms.** The $D_{98}$ and $D_{max}^{OAR}$ for the maximum and minimum scenarios as well as the deviation of $D_{98}$ and $D_{max}^{OAR}$—$\Delta D_{98}$ and $\Delta D_{max}^{OAR}$—over all reassessed dose distributions with 28 different perturbations of beam ranges and positions.

| Plan no | Phantom type | Plan name | $D_{98}$ nominal scenario | maximum scenario | minimum scenario | $\Delta D_{98}$ | $D_{max}^{OAR}$ nominal scenario | maximum scenario | minimum scenario | $\Delta D_{max}^{OAR}$ |
|---|---|---|---|---|---|---|---|---|---|---|
| 1 | Homogeneity | Conventional IMCT | 0.89 | 0.96 | 0.82 | 0.15 | 1.11 | 1.24 | 0.93 | 0.31 |
| 2 | | Range (Setup) + Heterogeneity robust IMCT | 0.98 | 0.98 | 0.81 | 0.17 | 0.96 | 1.16 | 0.91 | 0.25 |
| 3 | | Range (Setup) + Gradient robust IMCT | 0.90 | 0.94 | 0.83 | 0.11 | 1.01 | 1.15 | 0.91 | 0.24 |
| 1 | Heterogeneity | Conventional IMCT | 0.97 | 0.98 | 0.84 | 0.14 | 0.99 | 1.17 | 0.92 | 0.26 |
| 2 | | Range (Setup) + Heterogeneity robust IMCT | 0.96 | 0.98 | 0.81 | 0.18 | 0.99 | 1.16 | 0.96 | 0.20 |
| 3 | | Range (Setup) + Gradient robust IMCT | 0.98 | 0.98 | 0.87 | 0.11 | 0.94 | 1.09 | 0.90 | 0.19 |

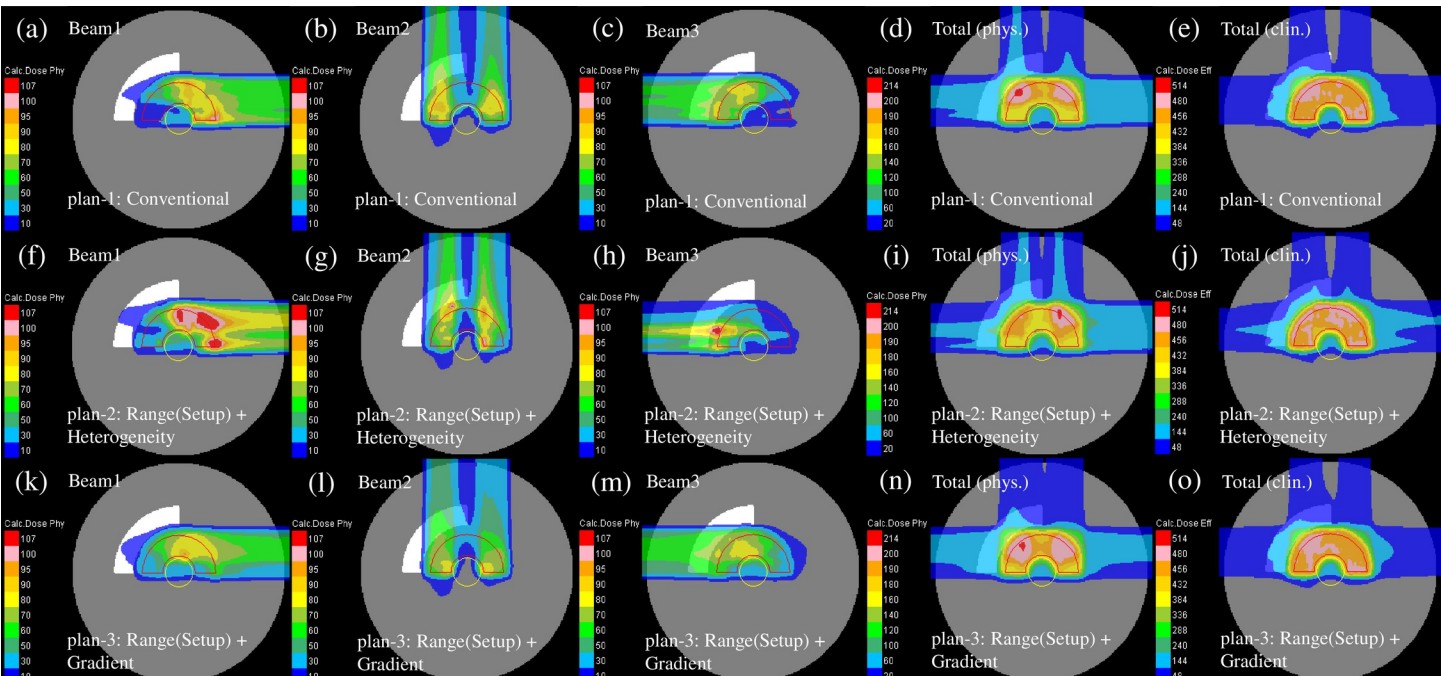

**Fig 8.** The individual dose distributions in the heterogeneous phantom of the beams with the port angles of 0˚ (the first column from the left), 90˚ (the second column from the left), and 270˚ (middle column) and the total physical dose distribution (the second column from the right) and clinical dose distribution (the first column from the right) for plan-1 (upper row), plan-2 (middle row) and plan-3 (lower low) optimized using the SC algorithm, respectively. The red and yellow lines demonstrate the CTV and OAR, respectively.

Comparable to the homogeneous phantom, the dose distribution of the WC IMCT demonstrated less deterioration among the plans.

The DVHs of the nominal and reassessed perturbed dose distributions are demonstrated with thick solid curves in Fig 12(C) and 12(D) and the hatched area in Fig 12(C) and 12(D). The robustness indicators of the CTV and OAR are depicted in Table 3. The outcomes were similar to those in the homogeneous phantom [Fig 7(A) and 7(B) and Table 3].

**Patient plan.** Computational duration of the optimization for the three treatment plans [i.e., conventional IMCT plan, range (setup) and gradient robust IMCT plan, and WC IMCT plan] on an HP Z840 Workstation with Intel Xeon CPU E5-2699 v4 @ 2.20GHz and 256 GB RAM were 3, 10, and 31 min, respectively. Computational duration for the dose calculation and robustness analysis were approximately 1 and 30 min, respectively.

The representative dose distribution for the nominal case of the conventional IMCT, range (setup) and gradient robust IMCT, and the WC IMCT plans is presented in Fig 15. As demonstrated in the phantom research, compared with the conventional IMCT, a robust radiobiological optimized IMCT plan revealed the area smeared out to flatten the dose distribution. However, the dose distributions of the individual beams in the range (setup) and gradient robust IMCT plan and WC IMCT plan were considerably distinct.

The DVHs of the nominal and recalculated perturbed dose distributions are indicated with thick solid curves in Fig 16(A)–16(C) and the hatched area in Fig 16(A)–16(C). The robustness indicators of the CTV and OAR are illustrated in Table 4. Compared with the conventional IMCT, the $\Delta D_{98}$ and $\Delta D_{max}^{OAR}$ were decreased by 2% for the WC IMCT and 2% for range (setup) and gradient robust IMCT. The $D_{98}$ in the minimum scenario was the highest and $D_{max}^{OAR}$ in the

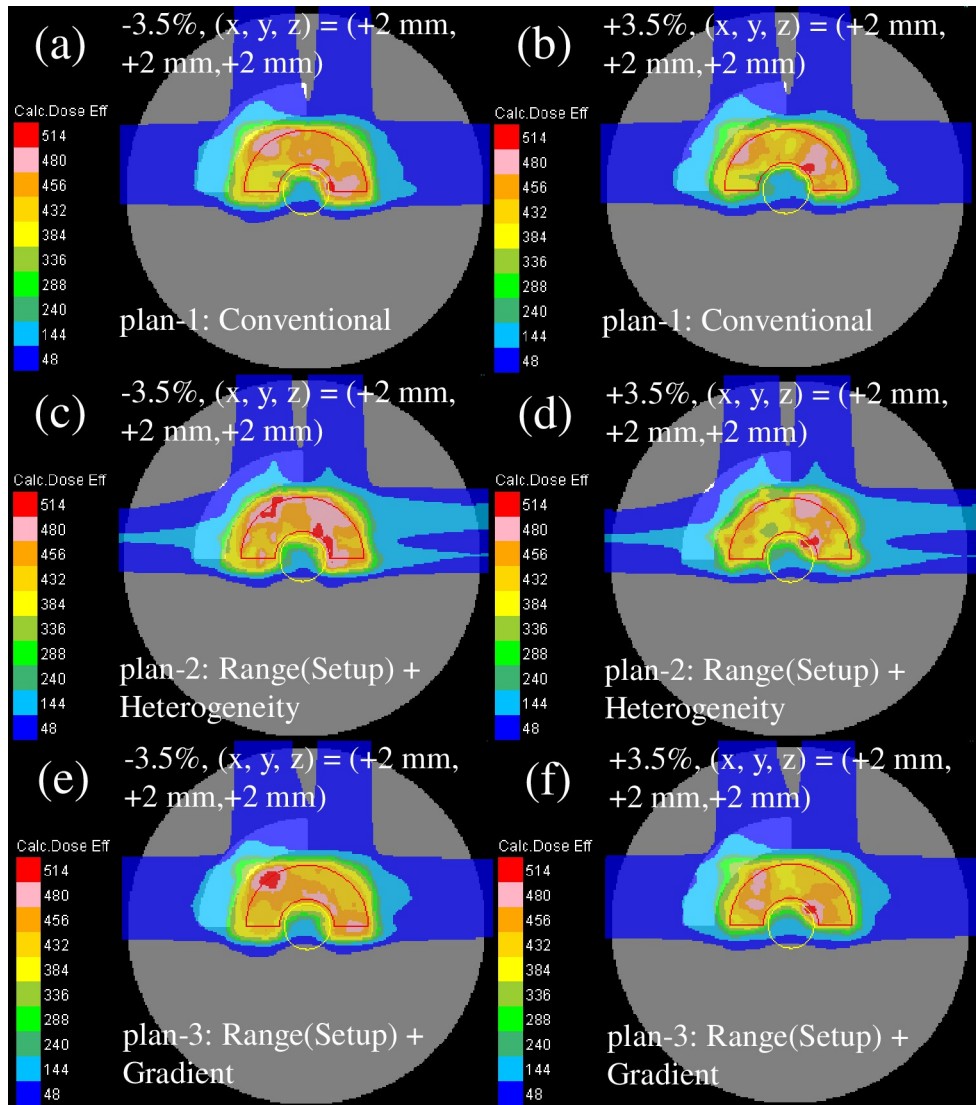

**Fig 9.** The total clinical dose distributions optimized using the SC algorithm in the heterogeneous phantom for plan-1 (upper row), plan-2 (middle row), and plan-3 (lower row) reassessed with the perturbations of the efficient density of −3.5% (left column) and +3.5% (right column) at the intentional translation of +2 mm in each direction for all fields. The red and yellow lines denote the CTV and OAR, respectively.

maximum was the lowest for the WC IMCT. The WC IMCT plan provided an enhanced clinical dose distribution with robustness.

## Discussion

In the present study, the computational efficacy of the newly developed IMCT algorithms was analyzed for the first time based on the mixed beam model in terms of physical and biological doses and were applied in the RTOG benchmark phantom under homogeneous and heterogeneous conditions and in a clinical case to verify the validity and efficacy of the robust radiobiological optimization. It was observed that the IMCT algorithm operated accurately, and the robust radiobiological optimization significantly decreased the range and position uncertainties in the inspected scenarios.

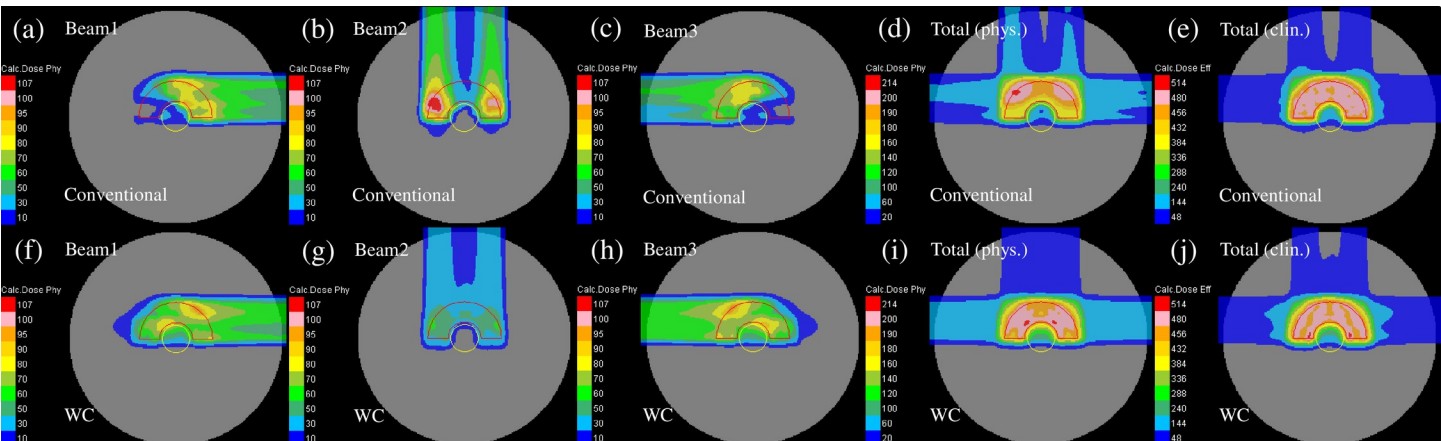

**Fig 10.** The individual dose distributions in the homogeneous phantom of the beams with the port angles of 0˚ (the first column from the left), 90˚ (the second column from the left), and 270˚ (middle column) as well as the total physical dose distribution (the second column from the right) and clinical dose distribution (the first column from the right) for conventional IMCT (upper row) and WC IMCT (lower low) optimized using the WC algorithm. The red and yellow lines denote the CTV and OAR, respectively.

The calculation method for RBE was one of the most essential variations among IMCT and SFUD in carbon-ion therapy. LQM parameters for carbon ions were determined considering the dose contributions from diverse fields that incorporate the IMCT separately from non-IMCT. Importantly, the mixed beam model was implemented as the RBE model for calculating

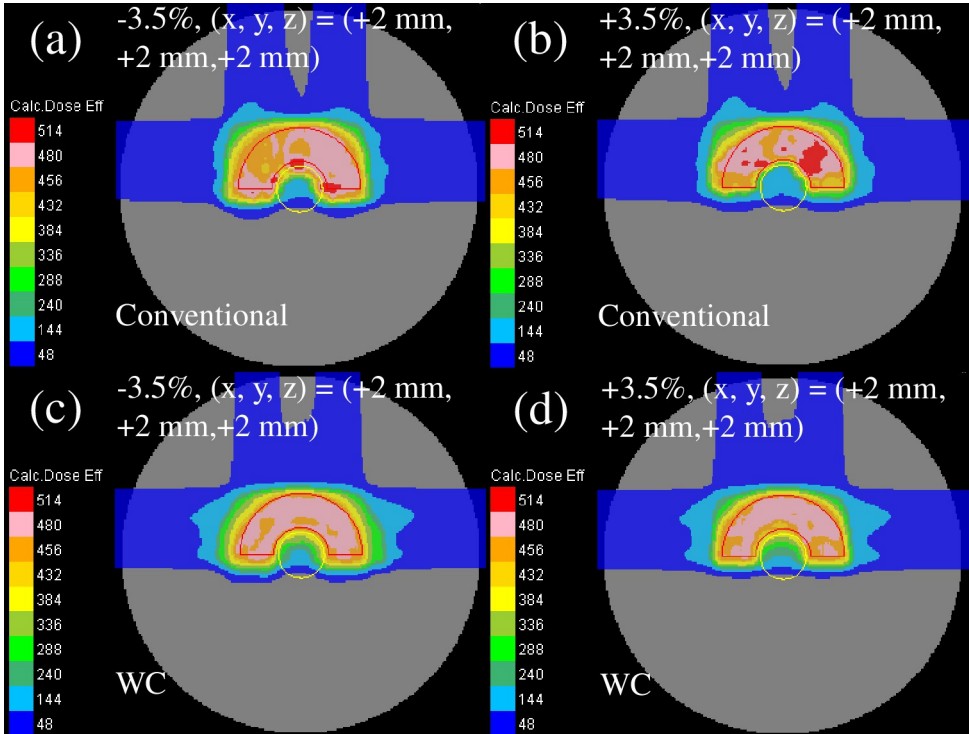

**Fig 11.** The total clinical dose distributions optimized using the WC algorithm in the homogeneous phantom for conventional IMCT (upper row) and WC IMCT (lower row) reassessed with the perturbations of effective density of −3.5% (left column) and +3.5% (right column) at the intentional translation of +2 mm in each direction for all fields. The red and yellow lines represent the CTV and OAR, respectively.

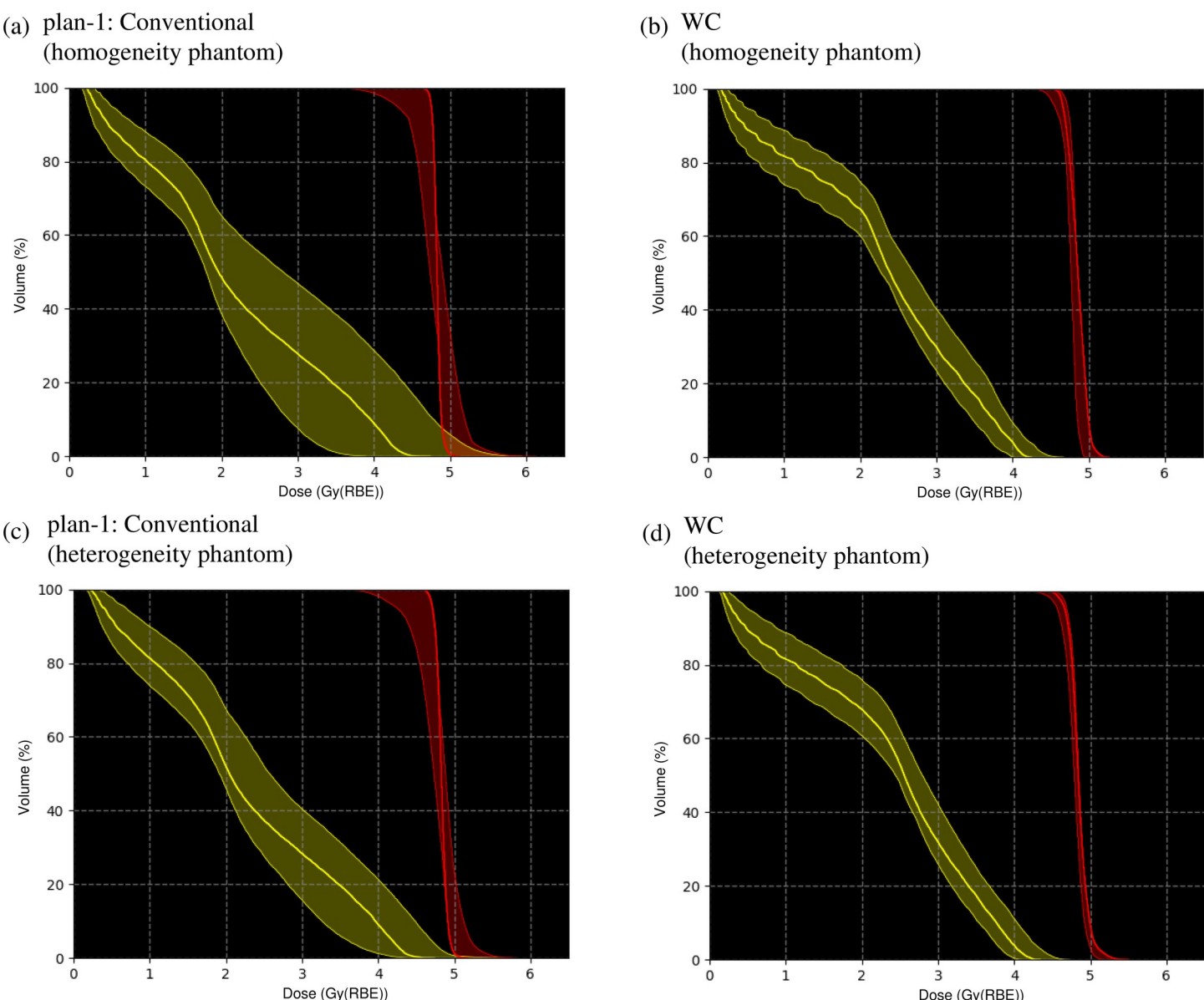

(a) plan-1: Conventional (homogeneity phantom)

(b) WC (homogeneity phantom)

(c) plan-1: Conventional (heterogeneity phantom)

(d) WC (heterogeneity phantom)

**Fig 12.** The variation of the DVHs of dose distributions optimized using the WC algorithm in homogeneous (upper row) and heterogeneous (lower row) phantom recalculated for 28 different perturbations of beam ranges and positions in the CTV and OAR for conventional IMCT and WC IMCT. The DVHs in the nominal case are represented by the thick solid line.

the RBE of the IMCT for the first time for the purpose of this study. Some previous studies have conducted the direct verification of the SF via experiment in the robust radiobiologically optimized IMCT. The independent validations were in agreement physically and biologically with the TPS, and the computational efficacy of the TPS for the IMCT was verified. The results suggest that the physical and biological dose distributions can be accurate in the error scenarios. In a previous study [6], it was reported that the small discrepancy between the computed outcomes and measurements was potentially attributed to a slight modification in the sensitivity of the used HSG cells. The systemic discrepancy itself is not problematic as long as the

**Table 3. The $D_{98}$ in the CTV and the $D_{max}^{OAR}$ for the nominal scenario optimized using the WC algorithm in homogeneous and heterogeneous phantom.** The $D_{98}$ and $D_{max}^{OAR}$ for the maximum and minimum scenarios and the deviation of $D_{98}$ and $D_{max}^{OAR}$—$\Delta D_{98}$ and $\Delta D_{max}^{OAR}$—over all reassessed dose distributions with 28 different perturbations of beam ranges and positions.

| Plan no | Phantom type | Plan name | $D_{98}$ nominal scenario | maximum scenario | minimum scenario | $\Delta D_{98}$ | $D_{max}^{OAR}$ nominal scenario | maximum scenario | minimum scenario | $\Delta D_{max}^{OAR}$ |
|---|---|---|---|---|---|---|---|---|---|---|
| 1 | Homogeneity | Conventional IMCT | 0.89 | 0.96 | 0.82 | 0.15 | 1.11 | 1.24 | 0.93 | 0.31 |
| | | WC IMCT | 0.94 | 0.96 | 0.92 | 0.04 | 0.94 | 1.01 | 0.88 | 0.12 |
| 1 | Heterogeneity | Conventional IMCT | 0.97 | 0.98 | 0.84 | 0.14 | 0.99 | 1.17 | 0.92 | 0.26 |
| | | WC IMCT | 0.96 | 0.97 | 0.93 | 0.04 | 0.90 | 0.99 | 0.85 | 0.13 |

similar difference is observed because of the slight modification in the sensitivity of the cell. A similar difference was also reported from the same research group [6].

In the present study, both SC and WC algorithms were utilized in the integrated TPS platform that can define the intensities of IMCT using scanned carbon-ion beams for robust radiobiological optimization. Among the SC algorithms evaluated, the Range (Setup) + Gradient robust IMCT decreased the deterioration experienced from range and position errors for the dose distribution regardless of the phantom type, as depicted in Figs 5 to 9 and Table 2. The current study revealed that the combination of the heterogeneity term was inefficient for adjusting spot weight for robustness enhancement. In the SC algorithm, the spot positions and weights were defined for the nominal scenario, and each spot weight was optimized to enhance robustness following the considered scenarios, thereby causing a steep dose gradient in each field comprising IMCT [Figs 5(F)–5(H) and 8(F)–8(H)] and decreasing the robustness in error scenarios. Furthermore, it was observed that the robust radiobiological optimization using the SC algorithm necessitated an in-field dose gradient suppression function to alleviate the dose distortion owing to the error scenarios; the margin adding to the CTV could thus be important. In another previous work [21], field-specific target volume was used to inflate the CTV; however, the beam-specific planning target volume (PTV) utilized in SFUD was deemed

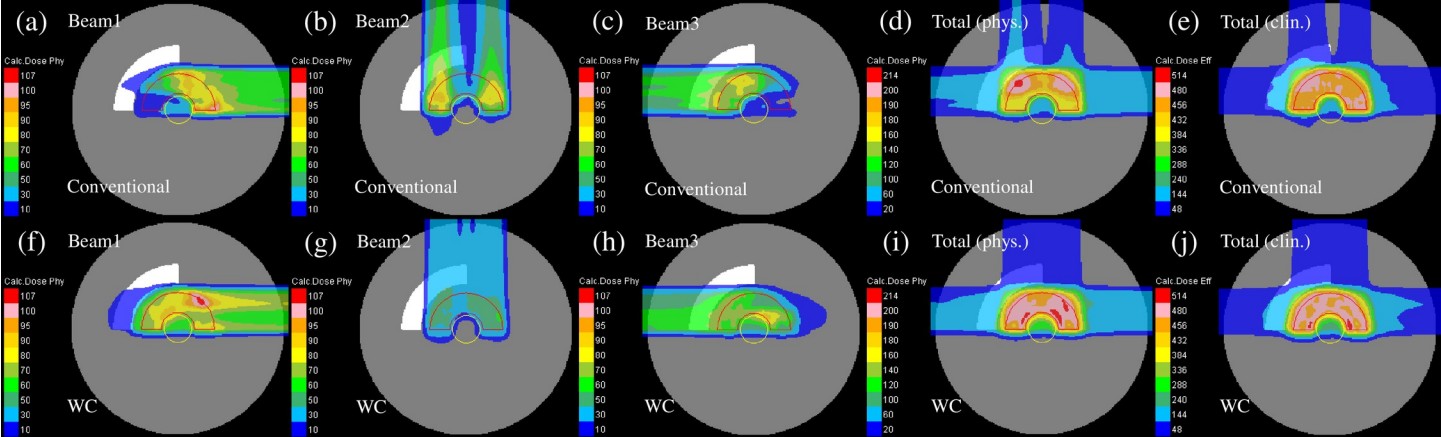

**Fig 13.** The individual dose distributions in the heterogeneous phantom of the beams with the port angles of 0˚ (the first column from the left), 90˚ (the second column from the left), and 270˚ (middle column) as well as a complete physical dose distribution (the second column from the right) and clinical dose distribution (the first column from the right) for conventional IMCT (upper row) and WC IMCT (lower low) optimized using the WC algorithm. The red and yellow lines denote the CTV and OAR, respectively.

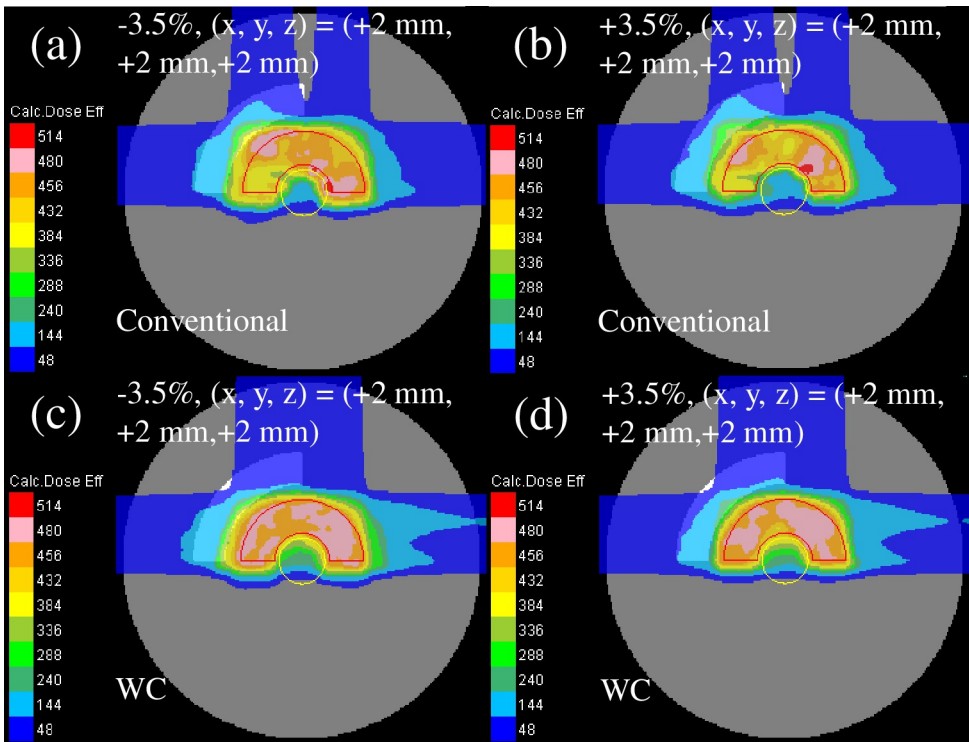

**Fig 14.** The total clinical dose distributions optimized with the WC algorithm in the heterogeneous phantom for conventional IMCT (upper row) and WC IMCT (lower row) reassessed with the perturbations of effective densities of −3.5% (left column) and +3.5% (right column) at the intentional translation of +2 mm in each direction for all fields. The red and yellow lines represent the CTV and OAR, respectively.

not suitable for the multifield optimized IMCT [31]. The priority of each term in the SC algorithm should be distinctly specified; otherwise, the spot weight adjustments would fail and the robustness would become susceptible to errors.

The WC IMCT lessened the uncertainties considerably as depicted in Figs 10–14 and Table 3. Typically, the dose distribution of individual fields is flattened in the beam and transverse directions when analyzing both range and setup errors irrespective of the employed robust optimization algorithms [16, 21]. Except for the SC algorithm, the WC algorithm automatically generates new spots based on the considered scenarios previously known as a "safety margin" [16]. In the present study, compared with the SC algorithms, the margin in WC algorithms could fortify the robustness of the improved dose distribution (Figs 9 vs. 14). As shown in Figs 15, 16, and Table 4, this occurrence was confirmed in the patient case, which was the first direct comparison between the two robust radiobiological optimization algorithms. Because the calculation was three times faster, the SC algorithm could be utilized for tumor sites that have less mobility. Typically, the computational load of the dose calculation for the carbon-ion beam was increased as the spot size of the carbon-ion beam is modest. Additionally, the WC algorithm could not be used owing to memory shortage and/or long computational duration in clinical practice for a bulky target. When using adequate geometrical margin, the SC algorithm attained robust radiobiological optimization for the case. The SC algorithm could be employed as an alternative to the WC IMCT in terms of computational costs.

Currently, there is no standard index for robustness evaluation. In the present study, the differences between the maximum and minimum of a DVH index in the evaluated scenarios

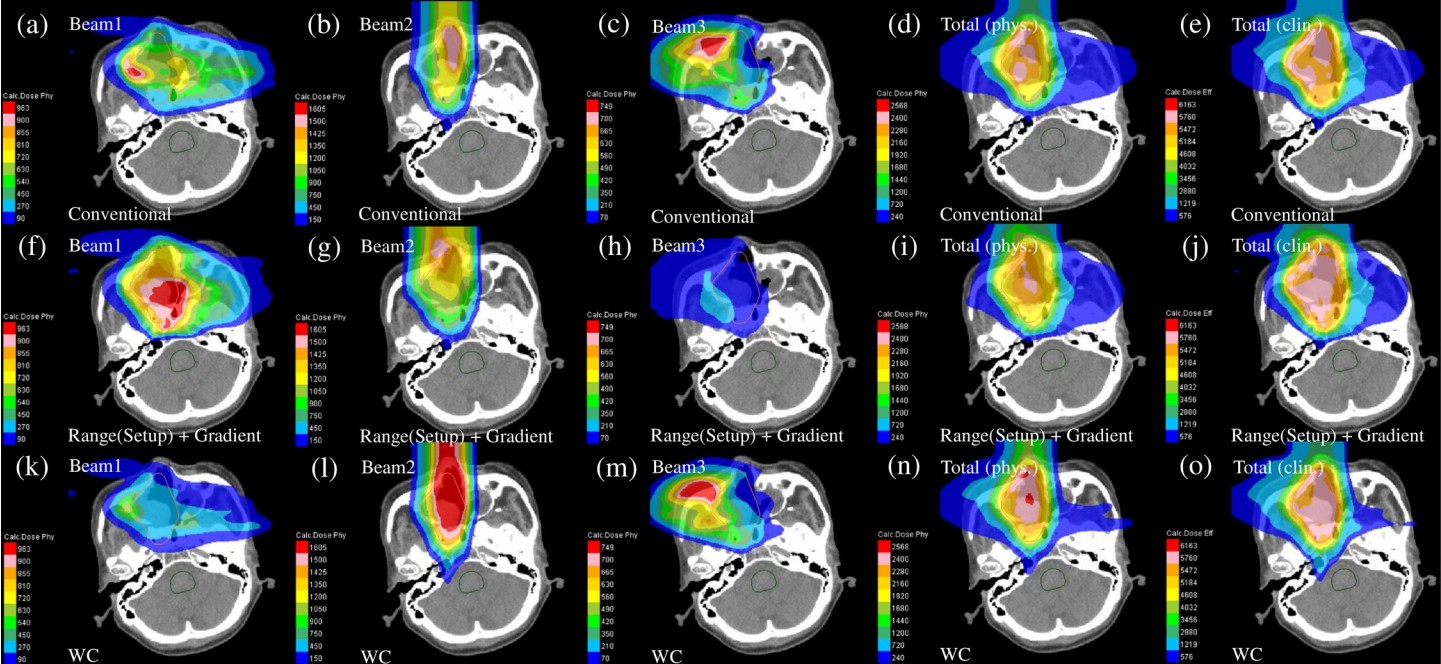

**Fig 15.** The individual dose distributions in the patient case of the beams with the port angles of 0˚ (the first column from the left), 90˚ (the second column from the left), and 270˚ (middle column) and the total physical dose distribution (the second column from the right) and clinical dose distribution (the first column from the right) for conventional IMCT (upper row), a range (setup) and gradient robust IMCT (middle row), and WC IMCT (lower low). The orange line depicts the CTV.

were used for robustness evaluation. Although several methods are presently employed for the robustness evaluation of the dose distribution (e.g., DVH index in WC scenario [32], DVH band [32], root mean square DVHs [31], and voxel-wise index [27]), only the voxel-wise index proved the correlation to (historic) PTV-based treatment plan evaluations. Nevertheless, the relationship between the applied approach and historic treatment plan evaluations required further investigation.

The dose-averaged LET is clinically used in carbon-ion therapy [9] though there are other physical quantities to describe beam quality such as lineal energy and/or specific energy which are distinguished based on whether or not considering stochastic nature of the carbon-ion

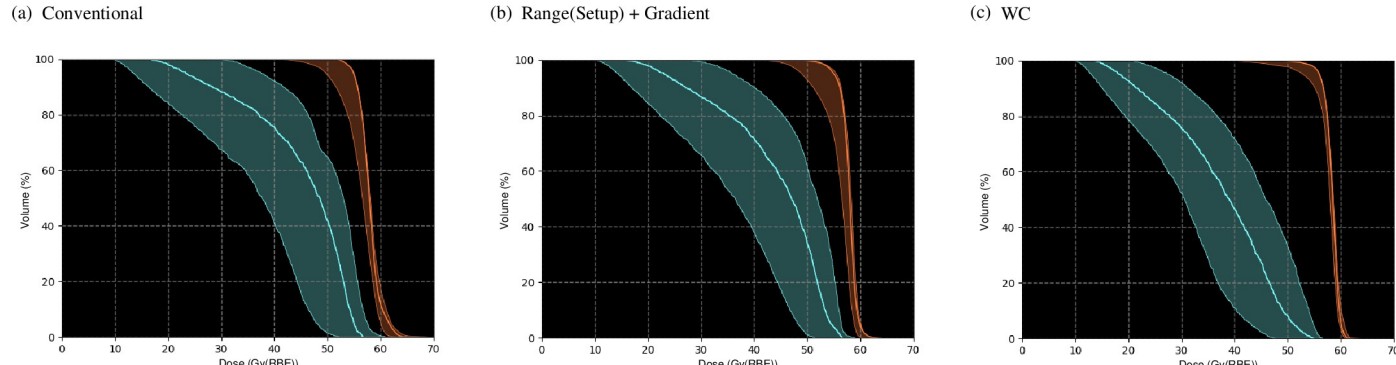

**Fig 16. The variation of the DVHs of dose distributions in the patient case reassessed for 28 different perturbations of beam ranges and positions in the CTV (orange) and OAR (light blue) for conventional IMCT, the range (setup) and gradient robust IMCT, and WC IMCT.** The thick solid line depicts the DVHs in the nominal case.

**Table 4. The $D_{98}$ in the CTV and the $D_{max}^{OAR}$ for the nominal scenario in the patient case.** The $D_{98}$ and $D_{max}^{OAR}$ for the maximum and minimum scenarios, and the deviation of $D_{98}$ and $D_{max}^{OAR}$—$\Delta D_{98}$ and $\Delta D_{max}^{OAR}$—over all reassessed dose distributions with 28 different perturbations of beam ranges and positions.

| | $D_{98}$ | | | | $D_{max}^{OAR}$ | | | |
|---|---|---|---|---|---|---|---|---|
| Plan Name | nominal scenario | maximum scenario | minimum scenario | $\Delta D_{98}$ | nominal scenario | maximum scenario | minimum scenario | $\Delta D_{max}^{OAR}$ |
| Conventional IMCT | 0.93 | 0.93 | 0.82 | 0.12 | 0.98 | 1.06 | 0.90 | 0.15 |
| Range(Setup) + Gradient robust IMCT | 0.93 | 0.93 | 0.81 | 0.12 | 0.98 | 1.02 | 0.89 | 0.13 |
| WC IMCT | 0.95 | 0.95 | 0.85 | 0.10 | 0.95 | 0.98 | 0.83 | 0.15 |

beam [33]. The present study did not evaluate the robustness of the LET distribution. However, the WC IMCT could inherently increase the robustness of the LET distribution owing to robust radiobiological optimization as depicted in Fig 13(C) and 13(D). In carbon-ion therapy, as RBE relies on LET, robust LET optimization would be of interest. A previous study [34] designed a LET-guided robust optimization (LETRO) considering the robustness of LET to directly optimize the LET distributions of the tumors and OARs explicitly in the robust optimization. The LETRO could be applied to further improve the robustness of the IMCT for the clinical dose distribution in combination with LET-painting [35, 36].

## Conclusions

In the present study, the computational efficacy of the newly developed IMCT algorithms was validated for the first time based on the mixed beam model in terms of physical and biological doses. Furthermore, both SC and WC algorithms were employed in the integrated TPS platform that can compute the intensities of IMCT using scanned carbon-ion beams for robust radiobiological optimization for the first time. The robust radiobiological optimizations significantly decreased the range and position uncertainties in the evaluated scenarios; the robustness of the WC algorithm was optimally compared with that of the SC algorithm. Nevertheless, the SC algorithm could be employed as an alternative to the WC IMCT considering its computational costs.

## Acknowledgments

The authors acknowledge and thank the staff at OHITC for their help with the measurements related to the commissioning, the staff at Osaka Heavy Ion Administration Company for help in operating the accelerator in the commissioning, Mr. Kenji Matsuda (Hitachi, Ltd. Smart Life Business Management Division) for supporting data analysis, and QA team in the Japan carbon-ion radiation oncology study group (J-CROS) and the QA committee in OHITC for a fruitful discussion on this work.

## Author Contributions

**Conceptualization:** Masashi Yagi, Toshiro Tsubouchi, Noriaki Hamatani, Masaaki Takashina, Shinichiro Fujitaka, Shusuke Hirayama, Hideaki Nihongi, Tatsuaki Kanai.

**Data curation:** Masashi Yagi, Toshiro Tsubouchi, Noriaki Hamatani, Masaaki Takashina, Naoto Saruwatari, Kazumasa Minami, Yushi Wakisaka, Shinichiro Fujitaka, Shusuke Hirayama, Hideaki Nihongi, Tatsuaki Kanai.

**Formal analysis:** Masashi Yagi, Toshiro Tsubouchi, Noriaki Hamatani, Masaaki Takashina, Naoto Saruwatari, Kazumasa Minami, Yushi Wakisaka, Shinichiro Fujitaka, Shusuke Hirayama, Hideaki Nihongi, Tatsuaki Kanai.

**Funding acquisition:** Masashi Yagi, Kazumasa Minami.

**Investigation:** Masashi Yagi, Toshiro Tsubouchi, Noriaki Hamatani, Masaaki Takashina, Naoto Saruwatari, Kazumasa Minami, Shinichiro Fujitaka, Shusuke Hirayama, Hideaki Nihongi, Tatsuaki Kanai.

**Methodology:** Masashi Yagi, Toshiro Tsubouchi, Noriaki Hamatani, Masaaki Takashina, Naoto Saruwatari, Kazumasa Minami, Yushi Wakisaka, Shinichiro Fujitaka, Shusuke Hirayama, Hideaki Nihongi, Tatsuaki Kanai.

**Software:** Masashi Yagi, Toshiro Tsubouchi, Noriaki Hamatani, Masaaki Takashina, Naoto Saruwatari, Yushi Wakisaka, Shinichiro Fujitaka, Shusuke Hirayama, Hideaki Nihongi.

**Supervision:** Azusa Hasegawa, Masahiko Koizumi, Shinichi Shimizu, Kazuhiko Ogawa, Tatsuaki Kanai.

**Validation:** Toshiro Tsubouchi, Noriaki Hamatani, Masaaki Takashina, Naoto Saruwatari, Kazumasa Minami, Yushi Wakisaka, Shinichiro Fujitaka, Shusuke Hirayama, Hideaki Nihongi.

**Visualization:** Masashi Yagi, Toshiro Tsubouchi, Noriaki Hamatani, Masaaki Takashina, Naoto Saruwatari, Kazumasa Minami, Yushi Wakisaka, Shinichiro Fujitaka, Shusuke Hirayama, Hideaki Nihongi.

**Writing – original draft:** Masashi Yagi.

**Writing – review & editing:** Masashi Yagi, Toshiro Tsubouchi, Noriaki Hamatani, Masaaki Takashina, Naoto Saruwatari, Kazumasa Minami, Yushi Wakisaka, Shinichiro Fujitaka, Shusuke Hirayama, Hideaki Nihongi, Azusa Hasegawa, Masahiko Koizumi, Shinichi Shimizu, Kazuhiko Ogawa, Tatsuaki Kanai.

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
