## [Decision Letter · Decision Letter 0]

17 Mar 2023

PONE-D-23-05244Development of robust radiobiological optimization algorithms based on the mixed beam model for intensity-modulated carbon-ion therapyPLOS ONE

Dear Dr. Masashi Yagi

Thank you for submitting your manuscript to PLOS ONE. After careful consideration, we feel that it has merit but does not fully meet PLOS ONE’s publication criteria as it currently stands. Therefore, we invite you to submit a revised version of the manuscript that addresses the points raised during the review process.

We look forward to receiving your revised manuscript.

Kind regards,

Xin A Wang, Ph.D.

Academic Editor

PLOS ONE

Journal Requirements:

"This work was partially supported by the JSPS KAKENHI 17K16437, 21K07700, and 22K07695. The authors acknowledge and thank the staff at OHITC for their help with the measurements related to the commissioning, the staff at Osaka Heavy Ion Administration Company for help in operating the accelerator in the commissioning, Mr. Kenji Matsuda (Hitachi, Ltd. Smart Life Business Management Division) for supporting data analysis, and QA team in the Japan carbon-ion radiation oncology study group (J-CROS) and the QA committee in OHITC for a fruitful discussion on this work."

"This work was partially supported by the JSPS KAKENHI 17K16437, 21K07700, and 22K07695."

Additional Editor Comments (if provided):

This manuscript evaluates some key features of a TPS in supporting IMCT. These include both the accuracy of physical and biologic dose calculation, plus the abilities of robust optimization. There are some valuable results such as the measurement of survival factions for in comparison to calculation. However, the manuscript is very difficult to read in current form. Overall, the manuscript is not written in a clear, consistent and coherency way. There are a lot of guessing on what the author is expressing. I strongly suggest the authors rewrite the manuscript with professional English editing.

Some specific comments include:

• It is unclear on the purpose of the manuscript. Most of work described in the manuscript is about validation of the TPS. Not much on the development except the description of SC and WC algorithms in the method and material section.

• Physical dose calculation validated by in house software. What makes the in house software the standard?

Reviewers' comments:

Reviewer's Responses to Questions

**Comments to the Author**

1. Is the manuscript technically sound, and do the data support the conclusions?

Reviewer #1: Yes

Reviewer #2: Yes

2. Has the statistical analysis been performed appropriately and rigorously? 

Reviewer #1: N/A

Reviewer #2: N/A

3. Have the authors made all data underlying the findings in their manuscript fully available?

Reviewer #1: Yes

Reviewer #2: Yes

4. Is the manuscript presented in an intelligible fashion and written in standard English?

Reviewer #1: Yes

Reviewer #2: Yes

5. Review Comments to the Author

Reviewer #1: The authors applied the mixed beam model in the physical and radiobiological dose optimization for intensity modulated carbon therapy (IMCT). Besides TPS, they have developed an independent dose calculation engine based on Python. Most importantly, they conducted cell irradiation to validated the accuracy of the TPS and independent dose engine. They found that the robust radiobiological optimization enhanced the sensitivity of the examined error by up to 19% compared to the conventional IMCT. Using the worst case scenario algorithm can even enhance the robustness more than the spot control algorithm. This is a very comprehensive study including TPS, independent dose verification engine, and cell irradiation experiments, phantom study, and optimization in patient case. I would recommend its acceptance for publication after minor revision. Nevertheless, I have to admit that this paper is really long and not easy to be understood especially for those who do not have experience in carbon therapy. Some comments are listed below:

(1) Page 8, line 37, it should be “surviving fraction”, not “survival fraction”. Please also change the label in figure 4.

(2) Page 8, line 47, what is “-a 0.6%”?

(3) page 16, line 279, in each plate. What type of cell culture plate did you use? Single-well petri dish?

(4) page 31, the last paragraph regarding the discussion of LET painting. I don’t think it is necessary to mention this because in carbon ion therapy, LET is no longer an appropriate quantity in plan optimization and evaluation. You have conducted the robustness analysis of radiobiological dose, which is the appropriate quantity. Because multiple secondary charged particles are generated in carbon therapy, the appropriate physical quantity is lineal energy and/or specific energy in microdosimetry, rather than LET.

(5) page 39, figure 4, there are results from experiment, experiment with SFUD and independent calculation, why isn’t there the surviving fraction predicted in your TPS by WC or SC algorithms?

Reviewer #2: This manuscript discusses an investigation of robust optimization strategies for intensity modulated carbon ion therapy (IMCT). The study included both physical uncertainties and biological doses using two optimization methods, spot control and worst case for testing phantom and patient cases. It also included both calculated biological effect and survival rate validation of a human salivary gland cell line. Both plan robustness and biological effect are important topics in clinical particle therapy currently and thorough investigations are needed. I think this manuscript should be of great interest for particle therapy researchers and practitioners. The authors did a good job in explaining the methodologies and providing detailed supporting data. However, I have several comments as summarized in below with an aim towards improving the clarity of the manuscript.

I would suggest the authors to consider revising the title of this study, as well as abstract and other discussions if related. First, it seems the robust biological optimization algorithms had been developed in previous studies or from existing TPS, e.g., spot control (SC) and worst case (WC) optimization algorithms, and RBE optimization based on local effect or MK models. I think “development” might not be appropriate in the title. Some options could be “evaluation” or “assessment”, etc. Secondly, the term “mixed beam” is not descriptive. It could mean a mix of photon and particle beams, or a mix of different beam lines, or different RBE models, without a definition. It should be stated clearly what “mixed” means in the title and its first appearance in the main text.

The measurement of survival factions for the selected cell line presented interesting results for readers. However, it was not discussed with more detail. It would be more helpful if the authors could comment on some important aspects, such as the correlation between calculated and measured results, any insights from the experiments, the progress concerning previous validation, the current approach, and maybe future works.

Some specific comments include:

• Line 54. It could be more appropriate to state “the impact of range and position uncertainties”, because robust optimization does not reduce uncertainties directly.

• Line 134. It may be not necessary to add “independently from non-IMCT” as it is inherent in IMCT that each field is independently modulated. Same for Line 148.

• Line 152. How are alpha_ij and beta_ij determined? Details or a previous publication could be useful.

• Line 220. Is there a typo in equation (16)? Terms in the bracket seem identical.

• Line 233. Is the “dose calculation engine” based on analytical model or Monte Carlo? Is there any validation work for it, maybe in another work?

6. PLOS authors have the option to publish the peer review history of their article (what does this mean?). If published, this will include your full peer review and any attached files.

Reviewer #1: No

Reviewer #2: No

---

## [Author Response · Author response to Decision Letter 0]

9 May 2023

Please look at the Response to Reviewers file uploaded to the submission system.

---

## [Decision Letter · Decision Letter 1]

23 Jun 2023

PONE-D-23-05244R1Development of robust radiobiological optimization algorithms based on the mixed beam model for intensity-modulated carbon-ion therapyPLOS ONE

Dear Dr. Yagi,

Thank you for submitting your manuscript to PLOS ONE. After careful consideration, we feel that it has merit but does not fully meet PLOS ONE’s publication criteria as it currently stands. Therefore, we invite you to submit a revised version of the manuscript that addresses the points raised during the review process.

We look forward to receiving your revised manuscript.

Kind regards,

Xin A Wang, Ph.D.

Academic Editor

PLOS ONE

Journal Requirements:

Additional Editor Comments:

This manuscript evaluates some key features of a TPS in supporting IMCT. These include both the accuracy of physical and biologic dose calculation, plus the abilities of robust optimization. The study provides some valuable results in the advancement of IMCT and is clearly worth the publication. Both reviewers feel the revision addresses their previous comments and concerns. I agree the revision has significant improvement from previous submission. However, there are still some concerns as listed below:

1. The revision is more readable. The abstract and introduction sections can be improved to make the manuscript appear in a clear, consistent and coherency way.

2. Strong recommendation on changing the title. The manuscript focused on the validation of the dose calculation of robust optimization algorithms. We expressed concern on using the world “development” which mismatch the purpose of the study.

3. Physical dose calculation validated by in house software. What makes the in house software the standard?

Reviewers' comments:

Reviewer's Responses to Questions

**Comments to the Author**

1. If the authors have adequately addressed your comments raised in a previous round of review and you feel that this manuscript is now acceptable for publication, you may indicate that here to bypass the “Comments to the Author” section, enter your conflict of interest statement in the “Confidential to Editor” section, and submit your "Accept" recommendation.

Reviewer #1: All comments have been addressed

Reviewer #2: All comments have been addressed

2. Is the manuscript technically sound, and do the data support the conclusions?

Reviewer #1: Partly

Reviewer #2: Yes

3. Has the statistical analysis been performed appropriately and rigorously? 

Reviewer #1: I Don't Know

Reviewer #2: N/A

4. Have the authors made all data underlying the findings in their manuscript fully available?

Reviewer #1: Yes

Reviewer #2: No

5. Is the manuscript presented in an intelligible fashion and written in standard English?

Reviewer #1: Yes

Reviewer #2: Yes

6. Review Comments to the Author

Reviewer #1: I don't have more questions. The authors have addressed all my concerns. I recommend its acceptance for publication.

Reviewer #2: (No Response)

7. PLOS authors have the option to publish the peer review history of their article (what does this mean?). If published, this will include your full peer review and any attached files.

Reviewer #1: No

Reviewer #2: No

---

## [Author Response · Author response to Decision Letter 1]

26 Jun 2023

Please see the uploaded document.

---

## [Editor Report · Decision Letter 2]

29 Jun 2023

Validation of robust radiobiological optimization algorithms based on the mixed beam model for intensity-modulated carbon-ion therapy

PONE-D-23-05244R2

Dear Dr. Yagi,

We’re pleased to inform you that your manuscript has been judged scientifically suitable for publication and will be formally accepted for publication once it meets all outstanding technical requirements.

Kind regards,

Xin A Wang, Ph.D.

Academic Editor

PLOS ONE

Additional Editor Comments (optional):

This manuscript evaluates some key features of a TPS in supporting IMCT. These include both the accuracy of physical and biologic dose calculation, plus the abilities of robust optimization. The study provides some valuable results in the advancement of IMCT and is clearly worth the publication.

The readability of manuscript was greatly improved through 2 revisions. It can be better but I feel it is publishable at present form. The new title really reflect the purpose of the study.
---

## [Editor Report · Acceptance letter]

20 Jul 2023

PONE-D-23-05244R2 

Validation of robust radiobiological optimization algorithms based on the mixed beam model for intensity-modulated carbon-ion therapy 

Dear Dr. Yagi:

I'm pleased to inform you that your manuscript has been deemed suitable for publication in PLOS ONE. Congratulations! Your manuscript is now with our production department. 

Kind regards, 

on behalf of

Dr. Xin A Wang 

Academic Editor

PLOS ONE